# REI-BENCH: CAN EMBODIED AGENTS UNDERSTAND VAGUE HUMAN INSTRUCTIONS IN TASK PLANNING?

**Chenxi Jiang    Chuhao Zhou    Jianfei Yang**[*]
MARS Lab, School of Mechanical and Aerospace Engineering
Nanyang Technological University
Singapore 639798
{chenxi003, chuhao002, jianfei.yang}@ntu.edu.sg

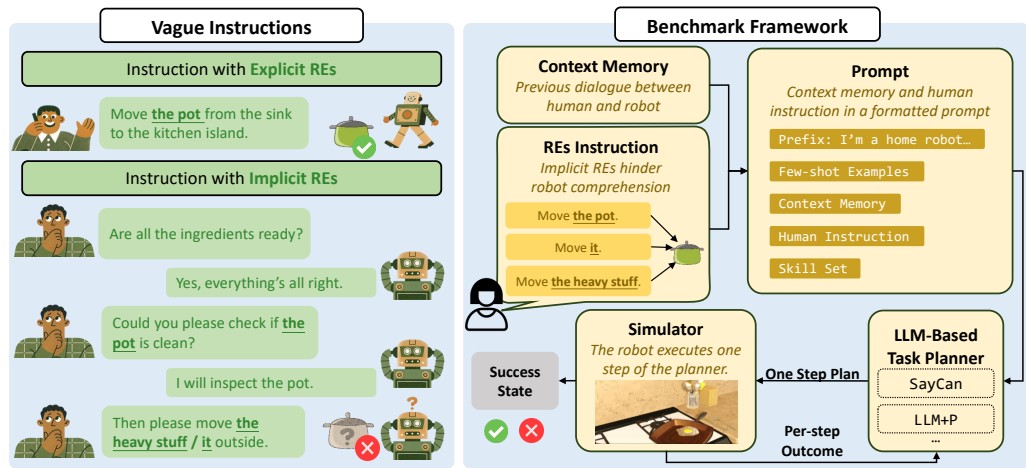

Figure 1: **Left**: Robots using existing LLM-based task planners can understand clear instructions with explicit referring expressions (REs), but they struggle to resolve implicit REs in multi-turn dialogues. **Right**: We propose the REI-Bench framework that aims to study real-world HRI scenarios where coreferential vagueness exists in human instructions.

## ABSTRACT

Robot task planning decomposes human instructions into executable action sequences that enable robots to complete a series of complex tasks. Although recent large language model (LLM)-based task planners achieve amazing performance, they assume that human instructions are clear and straightforward. However, real-world users are not experts, and their instructions to robots often contain significant vagueness. Linguists suggest that such vagueness frequently arises from referring expressions (REs), whose meanings depend heavily on dialogue context and environment. This vagueness is even more prevalent among the elderly and children, who are the groups that robots should serve more. This paper studies how such vagueness in REs within human instructions affects LLM-based robot task planning and how to overcome this issue. To this end, we propose the first robot task planning benchmark that systematically models vague REs grounded in pragmatic theory (REI-Bench), where we discover that the vagueness of REs can severely degrade robot planning performance, leading to success rate drops of up to 36.9%. We also observe that most failure cases stem from missing objects in planners. To mitigate the REs issue, we propose a simple yet effective approach: task-oriented context cognition, which generates clear instructions for robots, achieving state-of-the-art performance compared to aware prompts, chains of thought, and in-context learning. By tackling the overlooked issue of vagueness, this work contributes to the research community by advancing real-world task planning and making robots more accessible to non-expert users, e.g., the elderly and children.

[*]Corresponding author.
Project page and code are available at https://jcx0110.github.io/rei-bench-project

# 1 INTRODUCTION

In recent years, large language models (LLMs) have shown strong capabilities in tackling open-world tasks across diverse domains. Their use in robot planning indicates a promising shift: unlike traditional task planning methods (Hart et al., 1968; Aeronautiques et al., 1998; Chaslot et al., 2008), which are often constrained by specific environments and narrow task domains, LLMs enable robots to tackle tasks outside of conventional planning domains (Kim et al., 2024b; Zhou & Yang, 2025; Wu et al., 2024; An et al., 2024).

Although existing LLM-based task planners have shown remarkable performance (Ahn et al., 2022; Huang et al., 2022a; Wong et al., 2023), they rely on an idealized assumption: human instructions are always clear, complete, and unambiguous. However, in practice, human language often exhibits vagueness. Figure 1 shows how vague terms like "it" or "heavy stuff" (refer to a pan) can make a kitchen robot grab the wrong item (e.g., a plate or frying pan) instead of the pot. This challenge becomes even more pronounced for individuals with impaired memory or limited expressive abilities (Robinson & Apperly, 2001; So et al., 2010; Hendriks et al., 2014), including young children, older adults, and people with Alzheimer's disease—groups that rely most on robotic assistance.

Linguists suggest that various forms of linguistic vagueness can impact human-robot interaction, including syntactic, scopal, and coreferential vagueness (Li et al., 2024b). Among these, coreferential vagueness is particularly common and impactful in robot task planning, as it can affect the planner's understanding of noun phrases in instructions and lead to incorrect identification of task objects. The coreferential vagueness arises because humans employ not only explicit referring expressions (REs) (e.g., "pot"), which directly identify their referents, but also implicit REs (e.g., "it" or "this heavy thing"), which require contextual and environmental reasoning to resolve their meaning. Unlike robots, humans naturally use and interpret implicit REs in communication (Drave, 2002; Jucker et al., 2003; Paris et al., 2021; Peter, 2018; Alkhatnai, 2017; Wasow et al., 2005). Linguists have found that about 20% of expressions in news content are descriptive (a kind of implicit REs) (Hervaś & Finlayson, 2010), with this percentage being even higher in everyday life. Bridge inference theory (Clark, 1975) explains the process by which humans resolve implicit REs. When a listener hears an expression like "this heavy stuff," they naturally identify several possible referents based on contextual memory: the "pot," the "ingredients," and the "sink." Among these, the "pot" best matches the description. This explains why the word "refer" is composed of two parts: the prefix *re-* ("back/again") and the root *-fer* ("to carry / to bring"), which together embody the core mechanism of linguistic reference —namely, carrying meaning back from previously established contextual information. Inspired by linguistic studies, our research investigates key questions regarding task planners in embodied agents: **Do implicit REs in human instructions impact the performance of LLM-based task planners in robots? How does the frequency of implicit REs influence the success rate of these task planners? What are the underlying causes of this impact, and what strategies can be employed to mitigate it?**

To evaluate the impact of implicit REs on the success rate of planners, we first build the referring expressions instruction (REI) dataset and then propose the first benchmark that systematically models coreferential vagueness grounded in pragmatic theory for robot task planners, namely REI-Bench. In this benchmark, our work systematically models the use of REs in human-robot instructions by defining three levels of referential difficulty, based on the ratio of explicit to implicit REs. As robots need context to understand REs in human instruction, we propose three levels of real-world contexts in multi-turn dialogues with irrelevant or missing information. Combining REs' difficulties with context memory types, the REI dataset includes nine levels of coreferential vagueness.

Then we evaluate task planners based on the REI dataset, including 6 mainstream LLMs and 4 robot planning frameworks. Because most deployed robot systems currently rely on small-scale language models, our analysis focuses on planners operating within this model regime (Kim et al., 2024b). The results show that existing planners generally perform poorly in the presence of vagueness, with task success rates dropping between 7.4% and 36.9% in the baseline models. We attempt to mitigate the issue using novel basic NLP methods, including aware prompt (AP) (Gao et al., 2024a), chain-of-thought (CoT) (Wei et al., 2022), and in-context learning (ICL) (Brown et al., 2020), but observe limited gains. Meanwhile, we find that these performance declines in LLMs occur primarily because they devote excessive attention to plan generation while failing to perform their inherent language understanding abilities. This challenges the common assumption that simply embedding

an LLM in robot planning is sufficient for the robot to understand human language. Inspired by our observation, we propose a simple yet effective approach, Task-Oriented Context Cognition (TOCC), which decouples task comprehension from the planning decision-making process. Rather than designing a fully fledged solution, we intend to draw attention from the research community to this overlooked challenge and thereby motivate deep exploration.

Our contributions are threefold: (1) We systematically study how the instruction vagueness caused by REs impacts LLM-based robot task planners. To the best of our knowledge, this work provides the first systematic modeling of instruction vagueness in the context of robot task planning. (2) We develop REI-Bench by designing different levels of REs and context memory of human-robot dialogues, studying the success rate of robot tasks with vague instructions. (3) We analyze the underlying reasons for performance degradation and propose a simple yet effective approach, TOCC, to enhance the robustness of planners. Extensive experiments on REI-Bench validate its effectiveness.

## 2 RELATED WORKS

**Embodied Task Planning** enables robots to generate action sequences for various applications, including household and industrial tasks. Recent developments in LLM (Brown et al., 2020; Touvron et al., 2023) have led to language-driven planning methods (Kotb et al., 2024; Driess et al., 2023; Brohan et al., 2023; Chen et al., 2023; Liu et al., 2024a; Li et al., 2023; Huang et al., 2023), such as SayCan (Ahn et al., 2022), which leverage affordance functions to generate policies without fine-tuning. However, evaluating these planners becomes difficult due to the high cost of real-world deployment, including expensive robotic hardware and resources. The early studies rely on human evaluation (Ahn et al., 2022; Huang et al., 2022b), which was subjective and inefficient. To address this, automated evaluation with simulators has emerged (Yin et al., 2024; Liu et al., 2024b). Methods such as ProgPrompt (Singh et al., 2023) and LoTA-Bench (Choi et al., 2024) leverage datasets like VirtualHome (Puig et al., 2018) and AI2-THOR (Kolve et al., 2017). Meanwhile, Embodied Agent Interface (Li et al., 2024a) offers a general framework for evaluating LLMs.

**Linguistic Vagueness in LLMs** has attracted attention among researchers in natural language processing (Liu et al., 2023a; Ortega-Martín et al., 2023). The AmbiEnt benchmark (Liu et al., 2023a) evaluates the ability of LLMs to resolve linguistic ambiguities, revealing significant challenges even for advanced models such as GPT-4. To address these issues, APA (Kim et al., 2024a) improves the management of vague queries by LLMs by leveraging their self-assessment of vagueness. In the field of embodied AI, some prior works try to address these ambiguities by having robots ask clarifications or reason about uncertainty (Doğan et al., 2022; Park et al., 2023; Ren et al., 2023). However, these works lack a systematic definition of linguistic vagueness, a comprehensive evaluation of its impact on robotic performance, and effective methods to enhance the planner's own understanding capability. Building upon these developments, we further compare existing datasets and task planning benchmarks with linguistic ambiguity in Table 1.

Table 1: Comparison of REI-Bench with existing datasets and benchmarks.

| Benchmark / Method | Task | Planning | Systematic Vagueness | Multi-turn Context | Dataset / Size |
|---|---|---|---|---|---|
| **REI-Bench (ours)** | Evaluating coreferential vague instructions (RE-based) for robot task planning in AI2-THOR | ✓ | ✓ | ✓ | REI-Dataset / 2.7k instructions (× 9 vagueness levels) |
| AmbiK (Ivanova et al., 2025) | Ambiguous natural language task instructions for robot planning in a kitchen environment | ✗ | ✓ | ✗ | AmbiK / 1k ambiguous instructions |
| CLARA (Park et al., 2023) | Method for LLMs to classify whether the command is certain or not | ✗ | ✓ | ✗ | SaGC / 105 goals, 5222 tasks |
| KNOWNO (Ren et al., 2023) | Framework for measuring and aligning the uncertainty of LLM-based planners | ✓ | ✗ | ✗ | No dataset proposed |
| DialFRED (Gao et al., 2022) | Questioner performer framework | ✓ | ✗ | ✓ | 53K task-relevant questions and answers |
| Asking Clarifications (Xu et al., 2019) | Clarification identification, clarification question generation, and answering for ambiguous language | ✗ | ✓ | ✓ | CLAQUA / 40K dialogue words |

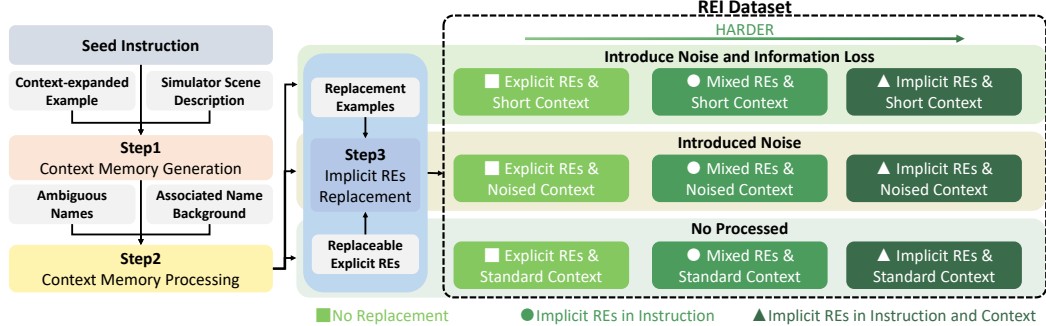

Figure 2: **Data curation pipeline of the REI dataset.** From a seed instruction, we (1) generate context memory; (2) produce three context variants—Standard, Noised, Short; (3) replace explicit REs with implicit ones across varying degrees. This results in subsets reflecting nine levels of coreferential vagueness, determined by RE types (Explicit/Mixed/Implicit) and context variants.

## 3 APPROACH

Existing works Choi et al. (2024) typically evaluate LLM-based robot task planners using clear instructions, whereas human instructions in real-world HRI scenarios are often ambiguous. In natural language, instruction vagueness arises from the one-to-many relationship between a *Signifier* (the symbol itself) and its *Signifieds* (the entities it may represent in the physical world) (Hervaś & Finlayson, 2010). For example, the signifier 'mouse' can refer to two signifieds: 'a rodent' or 'a computer input device'. Pragmatics scholar Levinson et al. (Levinson, 1983) further distinguish vagueness into two types: *Referential Expressions (REs)* and *Deictic Expressions (DEs)*. Humans can interpret REs through language context. For example, "Please bring me the mouse and keyboard" (referring to the computer input devices). In contrast, understanding DEs depends on environmental context, such as time, space, and the speaker's position, for example, "I want the book on the right side". Given our focus on LLM-based task planners, we confine our discussion to the impact of REs on task planning performance.

The objective of our work is to comprehensively evaluate and analyze how different levels of coreferential vagueness from implicit REs affect the planner's performance across diverse multi-turn dialogue contexts. To this end, we (1) systematically formalize REs in the HRI context (section 3.1), (2) establish the REI dataset and benchmark to evaluate planners on embodied tasks involving vague instructions (section 3.2), and (3) introduce a simple yet effective solution (section 3.3).

### 3.1 FORMALIZING VAGUENESS BY IMPLICIT RES AND HUMAN-ROBOT DIALOGUE CONTEXT

Levinson et al. (Levinson, 1983) propose that understanding humans' intention depends both on *Referring Expressions* and *Context Memory*, where context memory refers to the previous dialogues which provide hints to determine the unique meaning of REs. Here, we systematically model these two concepts into three levels to simulate varying degrees of vagueness.

**Referring Expressions.** In robot task planning, understanding REs is essential, as they provide key semantic cues for the goals of embodied tasks. Specifically, REs take various forms, including proper nouns (e.g., "apple"), definite noun phrases (e.g., "the apple"), indefinite noun phrases (e.g., "an apple"), pronouns (e.g., "it", "them"), and attributive expressions (e.g., "sweet fruit"). The first three forms, known as *explicit REs*, have only one potential corresponding object and can be directly understood. In contrast, the latter two forms, known as *implicit REs*, have multiple potential corresponding objects and should be identified by contextual inference. For example, the pronoun "it" simply indicates the referent is an object rather than a person, while "red fruit" only specifies the type and color of the item, without identifying which specific fruit it is.

To systematically model different forms of REs, we categorize them into three levels (Figure 2). At the "Explicit REs" level, all expressions are preserved as they appear in the original dataset. At the "Mixed" level, explicit REs in the instruction are replaced with implicit ones, while those in the context memory remain unchanged. The planner should refer to explicit REs to infer their implicit counterparts. At the "Implicit REs" level, all explicit REs are replaced with implicit ones, except the

first one in context memory, forcing the planner to rely on scene information to identify the referred objects. The latter two levels essentially simulate the vagueness introduced by implicit REs, which are more common in daily human communication.

**Context Memory.** Linguists argue that the connection between words and objects is constructed by humans within specific contexts (Levinson, 1983). Analogously, different types of context can affect the reasoning capability of a robot due to the inclusion of misleading cues or the omission of semantic information. One common source of misleading cues arises when a single signifier may plausibly refer to multiple, context-dependent entities, creating naming ambiguity. For example, the word "apple" may shift from denoting a fruit to referring to a mobile phone brand when such mentions appear in the discourse. Additionally, semantic omissions can arise from the speaker's health issues or from different stages of cognitive development.

To simulate these scenarios, we deliberately introduce irrelevant information to the context while removing certain cues. Specifically, we define three types of context memory. In the "standard context", all information related to the task in context is provided. In "Noised Context", we introduce the *Ambiguous Name* noise, defined as a character or brand with a name intentionally resembling that of an object in the simulator scene. For example, "Apple" as a brand name is repeatedly mentioned in the dialogue, causing the planner to treat the fruit in the scene as the target. This noise reflects real-world cases where names are the same as objects, and tests the model's ability to correctly identify the intended referent. Moreover, the "Short Context" not only introduces noise but also omits partial task-relevant information, further increasing the difficulty of reasoning.

## 3.2 REI-BENCH DATASET

Existing vague expressions datasets (Marcus et al., 2011; Levesque et al., 2012; Recasens et al., 2010), which are annotated by linguists, do not systematically formulate the position, frequency, and forms of REs, making it infeasible to investigate their impact on task planning. Thus, we establish a comprehensive referring expressions instruction (REI) benchmark via an automatic pipeline to assess the effects of implicit REs on robotic planning tasks.

Specifically, we build the REI-Bench dataset upon ALFRED (Shridhar et al., 2020), a benchmark for embodied household tasks. From ALFRED, we select six tasks (Pick & Place, Stack & Place, Clean & Place, Heat & Place, Cool & Place, Examine in Light) and exclude the task of Pick Two & Place as it cannot be reliably completed by an embodied agent. Furthermore, since tasks that cannot be accomplished even with clear instructions fall outside our scope, we use "LLaMA3.1-8B + SayCan" to perform the six selected tasks in the AI2-THOR (Kolve et al., 2017) simulator, keeping only the successfully executed tasks as seed instructions.

As shown in Figure 2, context memory is generated (Step 1), processed into three types (Step 2), and further transformed to the REI-Bench through implicit REs replacement (Step 3). In step 1, we extend the context of seed instructions by prompting GPT-4o-mini with a template that consists of a context-expanded example and the text-based simulator scene description. The generated data consists of an instruction and a corresponding context memory. Typically, the generated instruction conveys the same task requirement as the seed instruction but in a more human-like form. Meanwhile, the context memory captures the multi-turn human-robot dialogue before the instructions, which may include any objects present in the scene to reflect the complexity of real-world human dialogue.

In step 2, we construct three types of context. The "standard context" retains the context memory generated in step 1 without further processing. For the "Noised Context" type, LLM is prompted to repeatedly insert an ambiguous name into the dialogue without altering its meaning. The ambiguous name is derived by slightly modifying the name of a simulator object (e.g., Rose → Mrs. Rose). To ensure natural adaptation, the LLM is given a brief background prompt for the name (e.g., "Rose is a family member."). In the "short context" setting, partial noun phrases are randomly removed from the context, including those containing task-relevant explicit REs. In Step 3, explicit REs are replaced with implicit ones following the rules outlined in Section 3.1. We adopt a CoT approach to determine which explicit REs can be substituted. Substitution examples are selected from OntoNotes (Pradhan et al., 2013), ensuring that the implicit REs are consistent with natural language usage. To ensure consistency in the number of explicit REs across tasks, we define a counting-based rule for each level of implicit REs and discard any data that violates the rule. Consequently, the REI-Bench consists of 2,700 examples spanning nine difficulty levels, defined by combinations of varying RE vagueness and

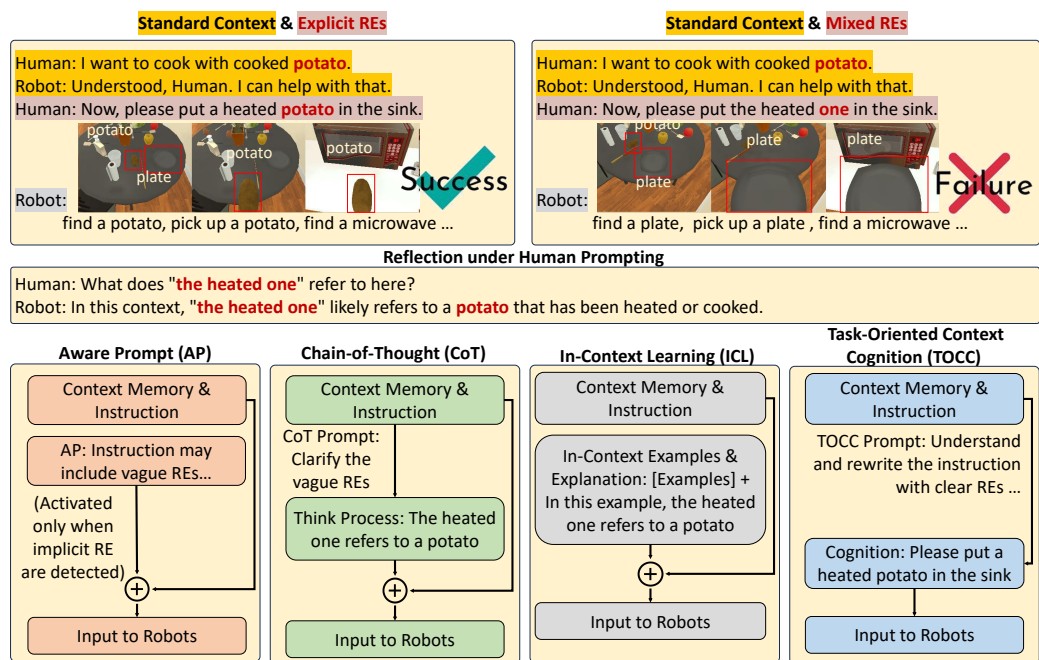

Figure 3: **Addressing implicit referring expressions in task planning.** Top row: LLM succeeds with explicit REs ("potato"), but misidentifies the object with implicit REs ("the heated one"). Middle row: a reflection prompt from humans can guide the LLM to resolve the implicit REs and identify the correct object. Bottom row: Comparison among different prompting methods, including aware prompt (AP), chain-of-thought (CoT), in-context learning (ICL), and our task-oriented context cognition (TOCC).

context memory conditions. Please refer to Appendix B for detailed prompts of context expansion (B.1), context process (B.2), explicit REs replacement (B.3), and the counting-based rule (B.4).

## 3.3 TASK-ORIENTED CONTEXT COGNITION

Based on the evaluation of multiple LLM-based robot planners in REI-Bench, we find that most failures result from object omissions, as illustrated in the top row of Figure 3. When explicit REs are presented in the instruction, the robot correctly identifies the task target "potato" and successfully completes the planning. In contrast, when implicit REs, such as "the heated one", are used, robots fail to identify "potato" but a task-irrelevant object "plate" instead. Furthermore, we find that LLMs can resolve implicit REs when prompted explicitly (shown in the middle row of Figure 3). This suggests that LLMs inherently can interpret implicit REs, yet this ability can not fully manifest during planning and requires explicit prompting. As a result, the idealized expectation that embedding an LLM into embodied agents guarantees full comprehension of human language has been challenged.

To this end, we propose to inject explicit prompts into the LLM-based robot planners to alleviate the coreferential vagueness. Specifically, we first evaluate three conventional prompting methods: (1) aware prompt (AP) (Gao et al., 2024a), which explicitly adds a prompt to guide the planner detect potential REs, (2) Chain-of-Thought (CoT) (Wei et al., 2022), which guides the planner to resolve REs step by step before planning, and (3) In-Context Learning (ICL) (Brown et al., 2020), which provides examples of how to infer implicit REs from context. However, AP remains insufficient for handling implicit REs, as the prompt signals vagueness but does not prompt deeper reasoning that LLMs struggle with during planning. Meanwhile, both CoT and ICL substantially lengthen prompts, hindering language understanding during planning, particularly when onboard computing resources are limited and only a small language model is available.

Consequently, we propose a simple yet effective mitigation strategy, Task-Oriented Context Cognition (TOCC), to tackle the challenges posed by REs. As shown in the bottom row of Figure 3, TOCC decouples the REs interpretation step from the planning process, avoiding the LLM from devoting excessive attention to planning within a single generation step. By resolving the vague REs and

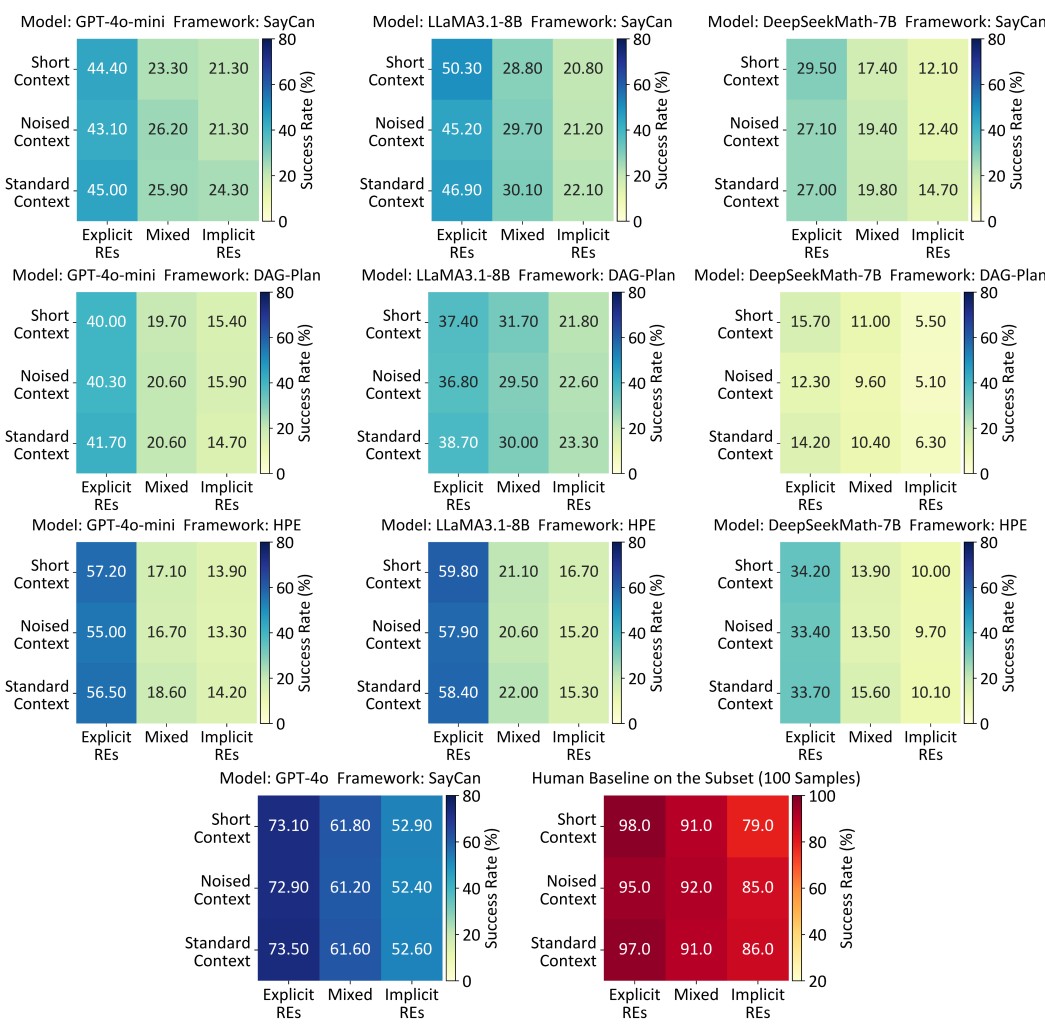

Figure 4: Success rate (%) of three task planner frameworks, SayCan, DAG-Plan, and HPE, using three LLMs (GPT-4o-mini, LLaMA3.1-8B, DeepSeekMath-7B), together with an additional "GPT-4o + SayCan" planner and a human baseline on the REI dataset. Explicit, Mixed, and Implicit REs denote three levels of implicit REs in human instructions, and Standard, Noised, and Short Contexts represent three context memory types.

rephrasing the human instruction in a more concise form before planning, TOCC demonstrates superior performance compared to existing methods. Implementation details and the exact variants used for AP and CoT are provided in Appendix C.

# 4 EXPERIMENT

## 4.1 EXPERIMENTAL SETUP

We evaluate four state-of-the-art LLM-based embodied task planning frameworks, SayCan (Ahn et al., 2022), DAG-Plan (Gao et al., 2024b), hierarchical task planning and execution (HPE) (Han et al., 2025), and LLM+P (Liu et al., 2023b) on our REI-Bench dataset. Due to the deployment requirements on mobile robots, which must be lightweight and open-source for on-robot adaptation, we focus solely on relatively small language models. For each task planner, we evaluate six LLMs, including GPT-4o-mini (Achiam et al., 2023), LLaMA3.1-8B (Grattafiori et al., 2024), Ministral-8B (Jiang et al., 2024), Gemma2-9B (Team et al., 2024), DeepSeek-Math-7B (Shao et al., 2024), and Qwen2.5-7B (Bai et al., 2023), which form a comprehensive benchmark consisting of 12 planners in total. To ensure balanced task coverage during evaluation, we first construct a 1,000-task subset

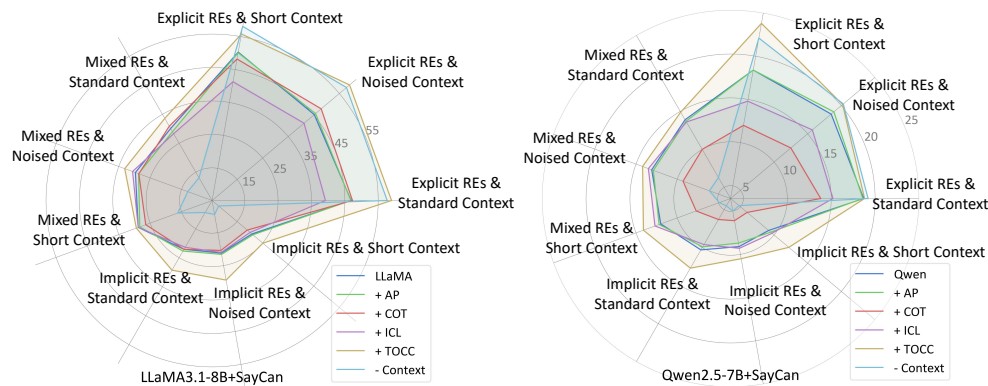

Figure 5: Success rates (%) of various prompting methods applied to LLaMA 3.1-8B and Qwen2.5-7B models with the SayCan framework on the REI dataset.

of REI-Bench via stratified sampling across task categories and use it for all planner evaluations. To further compare the performance of LLM-based task planners with humans, we invite human volunteers to conduct the same planning tasks on a randomly sampled subset of the REI-Bench.

## 4.2 BENCHMARK RESULTS OF LLM-BASED TASK PLANNERS

We evaluate the planning performance of all benchmark models on the REI dataset. Due to page limitations, we present only six benchmark results in Figure 4. Additional results and detailed comparisons are provided in Appendix A.3.

**LLM-based planners struggle to handle embodied tasks in multi-turn dialogue.** We use the instruction portion (excluding the context memory) of the "Explicit REs + Standard Context" type of data as a baseline, for which the "LLaMA3.1-8B+SayCan" planner achieves a 57.7% success rate. However, as shown in the middle of the top row in Figure 4, multi-turn dialogues in "Standard Context" cause the success rate of "LLaMA3.1-8B+SayCan" model to drop significantly from 57.7% to 46.90%, even without implicit REs. This performance gap highlights the limitations of existing LLM-based planners in handling natural, multi-turn human conversations.

**The performance of LLM-based planners consistently decreases as implicit REs increase, while remaining less affected by context memory noise.** With the increase of implicit REs, all benchmark planners demonstrate consistent performance degradation across "Standard", "Noised", and "Short" context memory. Take "LLaMA3.1-8B+SayCan" (middle of the top row in Figure 4) as an example, the success rate drops 16.8% / 15.5% / 21.5% at the "Mixed REs" level and further decreases 8.0% / 8.5% / 8.0 % at the "implicit REs" level. The consistent declines in LLMs' ability demonstrate that existing LLM-based planners cannot effectively handle the vagueness of implicit REs within human instructions for embodied tasks. In addition, compared to introducing multi-turn dialogues, adding naming ambiguity noise ("Noised Context") and further omitting partial task-relevant information ("Short Context") has little impact on performance. These observations suggest that existing LLM-based planners perform poorly when faced with multi-turn dialogues and implicit REs. However, such challenges are ubiquitous in human–robot interaction, especially when engaging with the elderly or children, and thus must be addressed.

## 4.3 ABLATION STUDY ON DIFFERENT PROMPTING METHODS

We compare four prompting methods that can mitigate the impact of implicit REs for LLM-based task planners: AP, CoT, ICL, and TOCC. As shown in Figure 5, by directly prompting the LLM-based planners that human instructions contain potential vagueness, AP improves performance in most scenarios. However, in some of the "Explicit REs", the performance decreases inversely. We deduce that these APs may lead the planner to hallucinate and detect vagueness even when instructions are clear. Meanwhile, ICL provides examples to help the planner infer the meaning of implicit REs from context. However, ICL leads to performance degradation in almost all categories. We deduce that the small onboarding LLM-based planners possess limited capability in learning from the

Figure 6: Failure example on "Mixed REs & Short Context", using LLaMA3.1-8B+SayCan. Due to the distracted object, the planner mistakenly put down the target object.

provided examples. Furthermore, CoT first guides the planner to autonomously analyse whether the instructions contain implicit REs and then perform planning based on the analysis, resulting in greater improvement. Building on the autonomous analysis from CoT, TOCC enables the planner to provide more refined and task-relevant instructions by correcting implicit REs and reorganizing language. Planning is then performed based on the enhanced instruction. By decoupling RE resolving from planning, TOCC obtains the best performance, with an average success rate improvement of 6.5% on "LLaMA3.1-8B+SayCan". We also provide results using only instructions as input. Under the Explicit REs type, the planner achieved performance comparable to TOCC. However, under the Mixed REs and Implicit REs types, planner performance dropped sharply. Compared to the original instructions, the enhanced instructions from our TOCC achieve improved planning performance. Moreover, the results are consistent with the pragmatic theory that context is indispensable for interpreting implicit REs. Please refer to appendix A.4 for completed ablation results.

## 4.4 ANALYSIS OF LLM-BASED PLANNER ERRORS

In this section, we review error cases made by LLM-based planners in processing implicit REs. As shown in Figure 6, the planner was uncertain whether "outer layer of the sandwich" referred to bread or lettuce, revealing a limitation of existing planners. Additional cases are provided in appendix A.6.

Table 2: Error rates (%) for the object omission and execution error types in different benchmark models. Results under the "Standard Context" for three types of implicit REs are reported.

| Model | Implicit REs Level | Error Type | | Overall Error Rate |
|---|---|---|---|---|
| | | Object Omission Rate | Execution Error Rate | |
| GPT-4o-mini | Explicit REs | 7.1 | 47.9 | 55.0 |
| | Mixed | 37.0 (+29.9) | 37.1 (−10.8) | 74.1 (+19.1) |
| | Implicit REs | 46.2 (+39.1) | 29.5 (−18.4) | 75.7 (+20.7) |
| LLaMA3.1-8B | Explicit REs | 22.6 | 30.5 | 53.1 |
| | Mixed | 38.8 (+16.2) | 31.1 (−0.6) | 69.9 (+16.8) |
| | Implicit REs | 53.9 (+31.3) | 24.0 (−6.5) | 77.9 (+24.8) |
| Deepseek-8B | Explicit REs | 28.6 | 44.4 | 73.0 |
| | Mixed | 40.8 (+12.2) | 39.4 (−5.0) | 80.2 (+7.2) |
| | Implicit REs | 57.8 (+29.2) | 27.5 (−16.9) | 85.3 (+12.3) |

For an in-depth analysis, we divide the task planning errors into two categories: object omission and execution error. Object omission refers to the planner not correctly identifying the target object in human instructions. As shown in Figure 3, the planner wrongly identifies the referring expression "one" to "plate" as a typical object. In addition, an execution error occurs when the planner identifies the correct object but cannot generate the complete sequence of actions to achieve the goal. We summarize the error rate related to object omission and execution error for different benchmark models in Table 2. For simplicity, only the results under the "Standard Context" are reported. Please

Table 3: Error rates (%) for the object omission and execution error types under different prompting methods (for LLaMA3.1-8B with "Standard Context").

| Method | Implicit REs Level | Error Type | | Overall Error Rate |
| | | Object Omission Rate | Execution Error Rate | |
|---|---|---|---|---|
| LLaMA3.1-8B | Explicit REs | 22.6 | 30.5 | 53.1 |
| | Mixed | 38.8 | 31.1 | 69.9 |
| | Implicit REs | 53.9 | 24.0 | 77.9 |
| + AP | Explicit REs | 22.7 (+0.1) | 30.5 | 53.2 (+0.1) |
| | Mixed | 31.3 (−7.5) | 39.7 | 71.0 (+1.1) |
| | Implicit REs | 49.9 (−4.0) | 27.4 | 77.3 (−0.6) |
| + CoT | Explicit REs | 22.5 (−0.1) | 30.2 | 52.7 (−0.4) |
| | Mixed | 34.9 (−3.9) | 34.2 | 69.1 (−0.8) |
| | Implicit REs | 47.6 (−6.3) | 30.3 | 77.9 (+0) |
| + ICL | Explicit REs | 22.7 (+0.1) | 38.1 | 60.8 (+7.7) |
| | Mixed | 32.7 (−6.1) | 39.0 | 71.7 (+1.8) |
| | Implicit REs | 49.9 (−4.0) | 28.7 | 78.6 (+0.7) |
| + TOCC | Explicit REs | **16.8** (−5.8) | 24.2 | 41.0 (−12.1) |
| | Mixed | **28.5** (−10.3) | 37.9 | 66.4 (−3.5) |
| | Implicit REs | **40.1** (−13.8) | 30.6 | 70.7 (−7.2) |
| - Context | Explicit REs | 17.2 (−5.4) | 25.1 | 42.3 (−10.8) |
| | Mixed | 81.6 (+42.8) | 5.3 | 86.9 (+17) |
| | Implicit REs | 85.1 (+31.2) | 5.5 | 90.6 (+12.7) |

refer to the appendix A.5 for more results under other context memory types. It can be seen that the overall error rate increases as the level of implicit REs grows. However, the error rates for object omission and execution error show opposite trends: the former increases significantly, while the latter decreases as the level of implicit REs grows. This indicates that the main cause of task planning errors is that the implicit REs induce the task planner to overlook target objects in HRI, thus making it unable to generate a task plan correctly. Furthermore, as shown in Table 3, our TOCC effectively reduces both the overall error rate and the object omission error rate across all three levels of implicit REs. These results demonstrate that TOCC effectively guides task planners to focus on target objects in human instructions, thereby enhancing robustness to finish instructions with vague REs.

## 5 CONCLUSIONS

We study how coreferential vagueness in human instruction affects robot task planning. By REI-Bench, we systematically simulate real-world language vagueness by categorizing REs and context memory. Extensive experiments show that implicit REs significantly reduce planning success rates. We explore the underlying reason and introduce the TOCC method, which effectively mitigates the negative effect of coreferential vagueness on robot task planner performance.

While our work discusses the impact of coreferential vagueness, human language vagueness is pervasive, and other forms of linguistic vagueness, such as deictic expressions, syntactic vagueness, and scopal vagueness, remain largely unexplored in the context of robot task planning. Furthermore, to isolate the effect of REs, we filter the dataset by selecting tasks that LLMs can complete under clear instructions. As a result, the dataset consists of simple, short-horizon, single-objective tasks. As future LLM-based planners become capable of solving more complex, clear-instruction tasks, we plan to extend our analysis to long-horizon scenarios. Moreover, experiments in the AI2-THOR simulator provide initial results of the planner's semantic understanding capabilities. They do not capture multimodal information, including visual and spatial perception, which is required for robots to interpret other types of vague instructions. Thus, our future work will focus on investigating the impact of deictic expressions on VLM-based task planners and validating the findings through experiments with physical robots.

ACKNOWLEDGMENTS

This work is supported by MOE Singapore Tier 1 Grant RG83/25, RS36/24, and a Start-up Grant from Nanyang Technological University.

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

APPENDICES

Within this supplementary material, we elaborate on the following aspects:

# A SUPPLEMENTARY EXPERIMENT RESULTS

## A.1 FRAMEWORK OF PLANNERS

**SayCans (Ahn et al., 2022)** is a method designed to help robots understand and carry out human instructions expressed in natural language. It breaks down a complex instruction into smaller steps suggested by the LLM, and then evaluates whether each step is possible in the real world with an affordance-based value function (a separate model trained on data about how robots interact with their environment). With the emergence of more capable language tools, we employ Guidance (Choi et al., 2024) to replace the affordance value function with LLM-based feasibility assessment, which allows selecting a skill in one generation pass and significantly reduces experiment time.

**DAG-Plan (Gao et al., 2024b)** is a planning framework in which an LLM generates a Directed Acyclic Graph (DAG) of sub-tasks rather than a linear sequence. Each node represents a symbolic high-level action, and edges denote dependency relations. This structure makes the plan explicitly hierarchical and ensures that prerequisite conditions are satisfied before execution. By modeling sub-task dependencies, DAG-Plan improves robustness on multi-object and multi-step tasks. As an official implementation of DAG-Plan has not yet been released, we re-implemented a DAG-Plan-style architecture based on the publicly available descriptions of its model structure and prompting strategy, and adapted it to our experimental setting.

**HPE (Han et al., 2025)** is proposed as a unified evaluation architecture for vision-language-action (VLA) models, introducing a hierarchical memory bank to ensure fair and structured long-horizon reasoning across different VLA systems. Although HPE is designed for embodied VLA models rather than LLM-based robotic task planners, we include it as a representative framework that incorporates hierarchical memory management. To enable comparison in our setting, we adapt the memory bank mechanism to an LLM-based robot task planner, following the publicly released implementation.

**LLM+P (Liu et al., 2023b)** is a hybrid framework that integrates large language models (LLMs) with classical symbolic planners to achieve robust and interpretable task planning from natural language instructions. The process typically involves three steps: First, the LLM translates high-level natural language commands into formal representations such as PDDL (Planning Domain Definition Language). Second, a classical planner, such as Fast Downward, computes a valid or optimal plan based on the generated problem and domain definitions. Finally, the LLM translated the resulting plan, a sequence of low-level actions, back into natural language, making it more interpretable for users.

## A.2 LANGUAGE MODELS LIST AND SAMPLED-SUBSET TASK-TYPE DISTRIBUTION

Table 4: List of language models used in the experiments. Model names are either from the OpenAI API or the HuggingFace model hub.

| Type | Model name | Model size | Remark |
|---|---|---|---|
| Closed-source | GPT-4o-mini | Unknown | |
| Open-source | LLaMA3.1-8B | 8B | Instruct |
| | Gemma2-9B | 9B | |
| | DeepSeekMath-7B | 7B | Instruct |
| | mistral-8B | 8B | Instruct |
| | Qwen2.5-7B | 7B | Instruct |

Table 5: Task-type distribution and average subtask steps in the 1,000-task sampled subset. The column "Original Proportion" corresponds to the original ALFRED dataset distribution, and our stratified sampling preserves this proportion. (GT denotes ground truth.)

| Task Type | Original Proportion (%) | Sampled Count | Avg. Subtask Steps (GT) |
|---|---|---|---|
| Cool & Place | 16.8% | 168 | 12 |
| Heat & Place | 16.8% | 168 | 14 |
| Clean & Place | 16.2% | 162 | 10 |
| Examine in Light | 13.3% | 133 | 4 |
| Stack & Place | 18.4% | 184 | 11 |
| Pick & Place | 18.5% | 185 | 9 |
| Pick Two (Excluded) | — | 0 | — |
| **Total / Average** | **100%** | **1000** | **10** |

The LLMs we used are listed in Table 4. To support the deployment of task planners on edge devices, we focus on lightweight, open-source language models with relatively small parameter sizes. For each model series, we use the latest available base version at the start of our study. Since a compact version of DeepSeek-v3 was not yet available, we use the experimental DeepSeekMath model instead in our evaluation.

## A.3 SUPPLEMENTARY BENCHMARK RESULTS

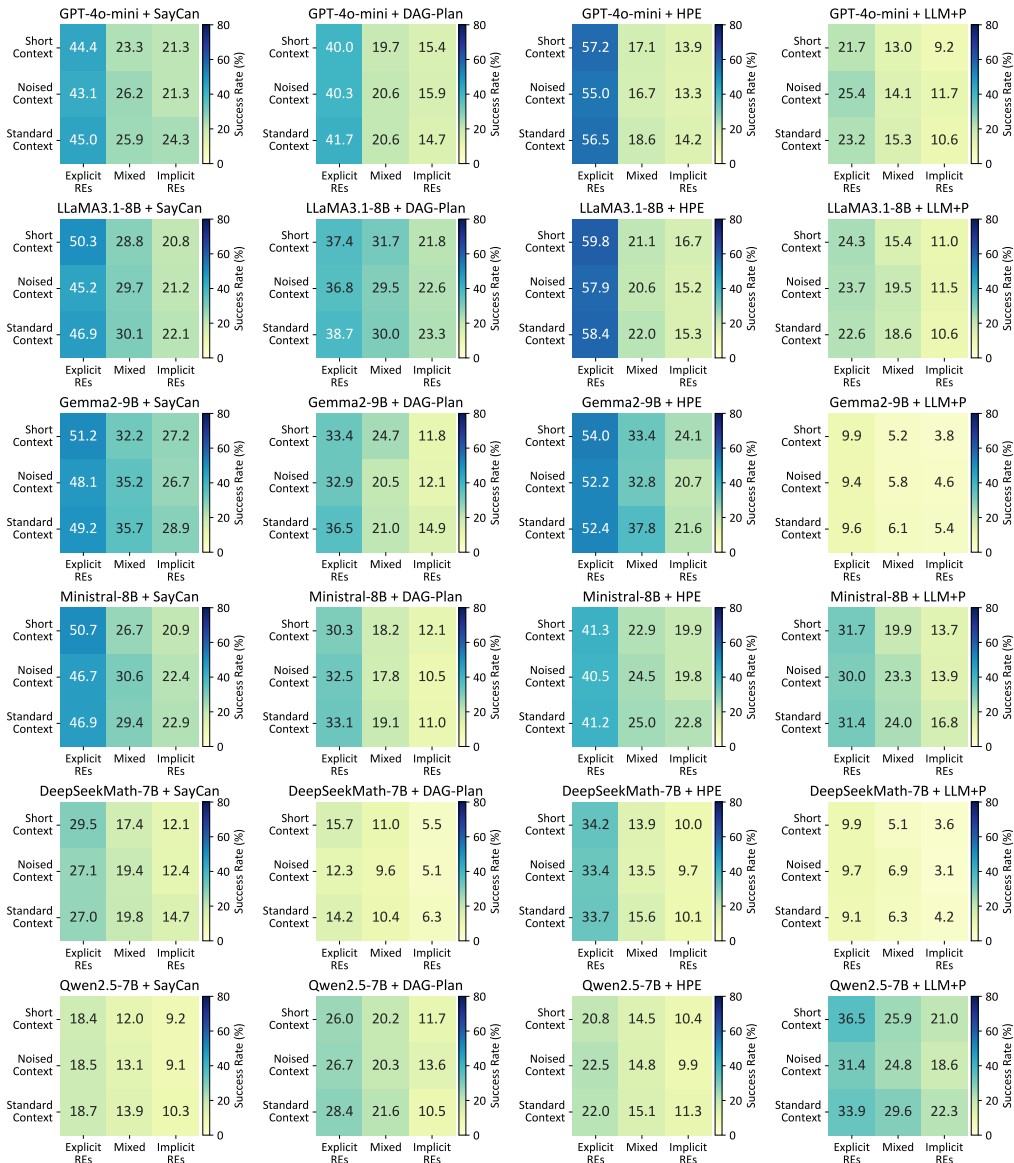

Figure 7: Success rate (%) of two task planner frameworks (SayCan, DAG-Plan, HPE, and LLM+P using three LLMs (GPT-4o-mini, LLaMA3.1-8B, DeepSeekMath-7B, Gemma2-9B, Ministral-8B, and Qwen2.5-7B) on REI dataset. Explicit, Mixed, and Implicit REs denote three levels of implicit REs in human instructions, and Standard, Noised, and Short Contexts represent three context memory types.

All experiments in this paper are conducted on task instances sampled from REI-Bench. As shown in Table 5, we construct a 1,000-task evaluation subset using stratified sampling that preserves the original ALFRED task-type distribution to prevent partial bias.

We observe that while modern planning paradigms introduce valuable structured memory or symbolic constraints, navigating implicit referring expressions (REs) remains a shared challenge. As shown in Figure 7, all evaluated planners exhibit two consistent trends: (i) a noticeable drop in success rates under multi-turn dialogue settings, and (ii) a steady degradation as the proportion of implicit REs increases.

Starting with the **SayCan** framework, performance varies across backbone LLMs but follows the same overall pattern: even the strongest variants (e.g., Gemma2-9B+SayCan) experience a marked decrease when explicit REs are replaced with implicit ones, underscoring the intrinsic difficulty of vague instruction understanding.

Frameworks incorporating explicit structural priors encounter similar trade-offs. For instance, **DAG-Plan** suffers performance drops when implicit REs are introduced. This suggests that while dependency-graph-based hierarchical planning provides clear execution structures, adapting it to capture the highly nuanced context required for vague reference interpretation remains non-trivial. Similarly, **HPE** leads to declines at the "Mixed REs" level, followed by further degradation under "Implicit REs." This indicates that although manually structured memory banks are effective for organizing information, their strict formatting might inadvertently filter out the subtle contextual cues necessary for resolving implicit REs.

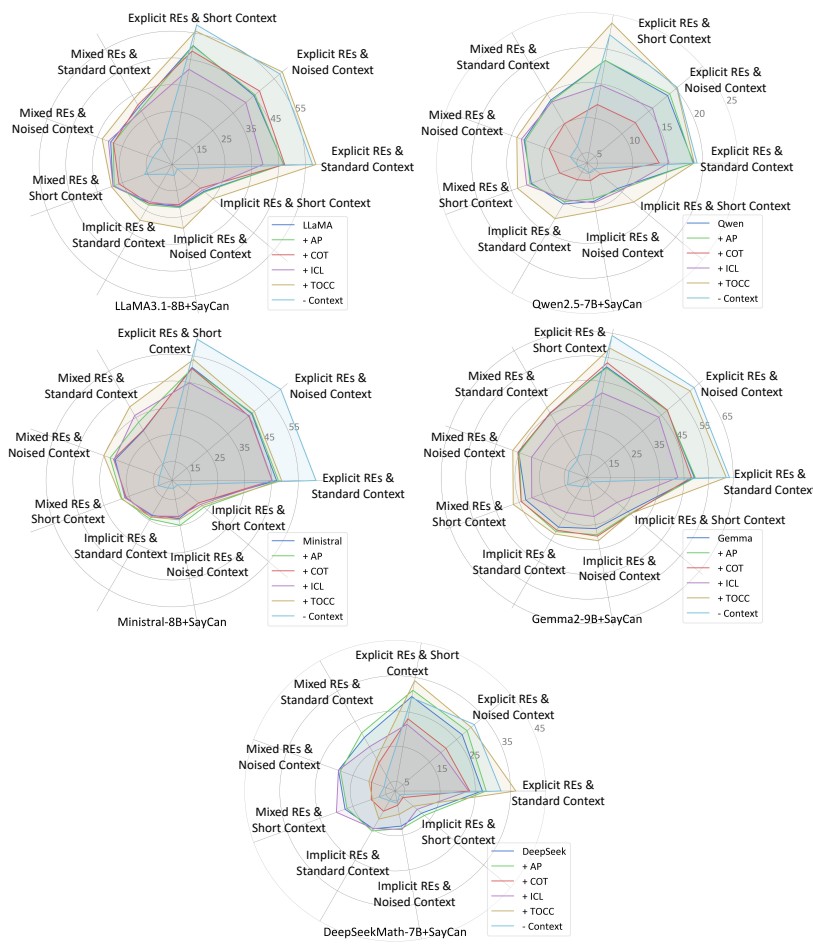

Figure 8: Success rates (%) of various prompting methods applied to LLaMA 3.1-8B, Gemma 2-9B, Ministral-8B, Qwen2.5-7B, and DeepSeekMath-7B models with SayCan framework on REI dataset.

Finally, **LLM+P**, which combines LLMs with a symbolic PDDL planner, typically performs lower than SayCan in these highly vague settings. While LLM+P has been shown to be highly effective for tasks with clear symbolic definitions, its reliance on a manually defined domain file and strict symbolic constraints presents challenges in multi-turn, context-dependent scenarios. Consequently, translating

the broad commonsense reasoning of LLMs into rigid symbolic states for implicit instructions proves difficult. Overall, the results demonstrate that the vagueness challenges highlighted by REI-Bench are not model-specific but constitute a systematic hurdle for existing LLM-based task planning frameworks.

## A.4 SUPPLEMENTARY RESULT OF ABLATION STUDY ON METHOD

Table 6: Average token usage and inference latency per planning step across all planning methods.

| Method | Avg Input Tokens | Avg Output Tokens | Avg Total Tokens | Latency (ms) |
|---|---|---|---|---|
| LLaMA3.1-8B + SayCan | 1822 | 45 | 1867 | 474.30 |
| AP | 1850 | 42 | 1892 | 492.75 |
| Gated AP | 1841 | 43 | 1884 | 487.07 |
| CoT | 3485 | 128 | 3613 | 917.92 (timeout occurred) |
| Short CoT | 3464 | 82 | 3546 | 705.64 |
| Segmented CoT | 2595 | 137 | 2732 | 946.36 (timeout occurred) |
| ICL (2-shot) | 3075 | 46 | 3121 | 514.65 |
| TOCC (ours) | 1894 | 62 | 1956 | 598.40 |

Figure 8 presents the success rates of five task planners after applying the three prompting methods: AP, CoT, and TOCC. "LLaMA3.1-8B+SayCan", "Gemma2-9B+SayCan", and "Ministral-8B+SayCan" follow the general trend observed earlier. TOCC consistently yields the best performance, followed by CoT and AP. However, two planners, Qwen2.5-7B+SayCan and DeepSeekMath-7B+SayCan, exhibit divergent behaviors. For Qwen2.5-7B+SayCan, although TOCC remains the most effective method, the application of CoT leads to a substantial performance drop. This may be due to Qwen2.5-7B's limited ability to follow multi-step reasoning instructions embedded in CoT-style prompts. In the case of DeepSeekMath-7B+SayCan, both CoT and TOCC result in a decline in performance. This is likely because DeepSeekMath-7B is an experimental model specifically trained for mathematical problem solving and has not undergone alignment with human preferences. Consequently, it exhibits the strongest hallucination tendencies and the weakest instruction-following ability among all evaluated models. Given its similarly poor performance in the baseline planning tasks, whether DeepSeekMath-7B is capable of supporting embodied intelligence remains questionable.

Table 6 further shows that TOCC introduces only a modest overhead relative to lightweight baselines while remaining substantially more efficient than CoT and ICL. Specifically, TOCC increases total token usage and latency by only 3.95% and 26.18% over the vanilla model, and by 2.38% and 22.87% over AP, respectively. In contrast, TOCC reduces token usage and latency by 45.32% and 15.20% compared with Short CoT, and lowers token consumption by 38.41% relative to standard ICL.

## A.5 Supplementary Result of LLM-based Planner Errors

Table 7: Error rates (%) for the object omission and execution error types in different benchmark models.

| Model | Context Memory Type | Implicit REs Level | Error Type | | Overall Error Rate |
|---|---|---|---|---|---|
| | | | Object Omission Rate | Execution Error Rate | |
| GPT-4o-mini | Standard | Explicit REs | 7.1 | 47.9 | 55.0 |
| | | Mixed | 37.0 | 37.1 | 74.1 |
| | | Implicit REs | 46.2 | 29.5 | 75.7 |
| | Noised | Explicit REs | 7.5 | 49.4 | 56.9 |
| | | Mixed | 36.2 | 37.6 | 73.8 |
| | | Implicit REs | 50.6 | 28.1 | 78.7 |
| | Short | Explicit REs | 8.7 | 47.3 | 56.0 |
| | | Mixed | 47.5 | 29.2 | 76.7 |
| | | Implicit REs | 53.4 | 25.3 | 78.7 |
| LLaMA3.1-8B | Standard | Explicit REs | 22.6 | 30.5 | 53.1 |
| | | Mixed | 38.8 | 31.1 | 69.9 |
| | | Implicit REs | 53.9 | 24.0 | 77.9 |
| | Noised | Explicit REs | 23.8 | 31.0 | 54.8 |
| | | Mixed | 39.3 | 31.0 | 70.3 |
| | | Implicit REs | 53.9 | 24.9 | 78.8 |
| | Short | Explicit REs | 22.6 | 27.1 | 49.7 |
| | | Mixed | 45.4 | 25.8 | 71.2 |
| | | Implicit REs | 57.9 | 21.3 | 79.2 |
| Deepseek-8B | Standard | Explicit REs | 28.6 | 44.4 | 73.0 |
| | | Mixed | 40.8 | 39.4 | 80.2 |
| | | Implicit REs | 57.8 | 27.5 | 85.3 |
| | Noised | Explicit REs | 31.1 | 41.8 | 72.9 |
| | | Mixed | 44.9 | 35.7 | 80.6 |
| | | Implicit REs | 61.6 | 26.0 | 87.6 |
| | Short | Explicit REs | 29.2 | 41.3 | 70.5 |
| | | Mixed | 51.6 | 31.0 | 82.6 |
| | | Implicit REs | 61.9 | 26.0 | 87.9 |

Table 8: Error rates (%) for the object omission and execution error types under different prompting methods.

| Method | Context Memory Type | Implicit REs Level | Error Type | | Overall Error Rate |
| | | | Object Omission Rate | Execution Error Rate | |
|---|---|---|---|---|---|
| LLaMA3.1-8B | Standard | Explicit REs | 22.6 | 30.5 | 53.1 |
| | | Mixed | 38.8 | 31.1 | 69.9 |
| | | Implicit REs | 53.9 | 24.0 | 77.9 |
| | Noised | Explicit REs | 23.8 | 31.0 | 54.8 |
| | | Mixed | 39.3 | 31.0 | 70.3 |
| | | Implicit REs | 53.9 | 24.9 | 78.8 |
| | Short | Explicit REs | 22.6 | 27.1 | 49.7 |
| | | Mixed | 45.4 | 25.8 | 71.2 |
| | | Implicit REs | 57.9 | 21.3 | 79.2 |
| + AP | Standard | Explicit REs | 21.4 | 38.6 | 60.0 |
| | | Mixed | 31.7 | 39.3 | 71.0 |
| | | Implicit REs | 50.4 | 27.0 | 77.4 |
| | Noised | Explicit REs | 21.1 | 37.3 | 58.4 |
| | | Mixed | 32.0 | 37.6 | 69.6 |
| | | Implicit REs | 50.7 | 26.9 | 77.6 |
| | Short | Explicit REs | 19.5 | 35.4 | 54.9 |
| | | Mixed | 39.9 | 31.9 | 71.8 |
| | | Implicit REs | 51.9 | 25.6 | 77.5 |
| + CoT | Standard | Explicit REs | 22.5 | 30.2 | 52.7 |
| | | Mixed | 34.9 | 34.2 | 69.1 |
| | | Implicit REs | 47.6 | 30.3 | 77.9 |
| | Noised | Explicit REs | 22.4 | 29.7 | 52.1 |
| | | Mixed | 35.2 | 34.7 | 69.9 |
| | | Implicit REs | 48.5 | 29.9 | 78.4 |
| | Short | Explicit REs | 22.9 | 28.9 | 51.8 |
| | | Mixed | 41.4 | 31.2 | 72.6 |
| | | Implicit REs | 51.8 | 27.2 | 79.0 |
| + ICL | Standard | Explicit REs | 22.7 | 38.1 | 60.8 |
| | | Mixed | 32.7 | 39.0 | 71.7 |
| | | Implicit REs | 49.9 | 28.7 | 78.6 |
| | Noised | Explicit REs | 21.7 | 37.1 | 58.8 |
| | | Mixed | 30.6 | 38.8 | 69.4 |
| | | Implicit REs | 50.5 | 28.5 | 79.0 |
| | Short | Explicit REs | 20.8 | 37.9 | 58.7 |
| | | Mixed | 33.4 | 37.8 | 71.2 |
| | | Implicit REs | 51.7 | 28.2 | 79.9 |
| + TOCC | Standard | Explicit REs | 16.8 | 24.2 | 41.0 |
| | | Mixed | 28.5 | 37.9 | 66.4 |
| | | Implicit REs | 40.1 | 30.6 | 70.7 |
| | Noised | Explicit REs | 16.1 | 24.9 | 41.0 |
| | | Mixed | 28.4 | 38.5 | 66.9 |
| | | Implicit REs | 42.9 | 27.7 | 70.6 |
| | Short | Explicit REs | 17.1 | 27.1 | 44.2 |
| | | Mixed | 22.8 | 46.8 | 69.6 |
| | | Implicit REs | 43.5 | 31.4 | 74.9 |
| - Context | Standard | Explicit REs | 17.2 | 25.1 | 42.3 |
| | | Mixed | 81.6 | 5.3 | 86.9 |
| | | Implicit REs | 85.1 | 5.5 | 90.6 |
| | Noised | Explicit REs | 16.8 | 25.4 | 42.2 |
| | | Mixed | 82.2 | 4.8 | 87.0 |
| | | Implicit REs | 84.3 | 6.2 | 90.5 |
| | Short | Explicit REs | 17.5 | 24.4 | 41.9 |
| | | Mixed | 80.2 | 3.6 | 83.8 |
| | | Implicit REs | 87.6 | 4.6 | 92.2 |

## A.6 SUPPLEMENTARY OF TASK PLANNING CASES

Human previous inquiry(Not Required to Execute): Hey there, can you help me with something? I want to make the room feel cozier, so I was thinking about adjusting the brightness of the lamp.
Human previous inquiry(Not Required to Execute): Yes, please do that! I want to make sure the lamp is set just right for reading. I also need to find a good pen for my notes.
Human previous inquiry(Not Required to Execute): I usually like blue ink, but any pen will do for now. I just want to make sure the lamp is bright enough to see my notes clearly.
Human previous inquiry(Not Required to Execute): That sounds perfect! I want to ensure I can see everything well. Now, can you also make sure the pen is within reach while I focus on my notes?
Human pending instruction: No, I think that covers everything. Please turn on a lamp while holding a pen.

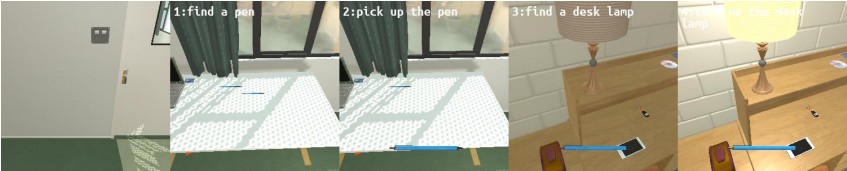

Human previous inquiry(Not Required to Execute): Hey there, can you help me with something? I want to make the room feel cozier, so I was thinking about adjusting the brightness of the lamp.
Human previous inquiry(Not Required to Execute): Yes, please do that! I want to make sure the lighting is set just right for reading. I also need to find a good writing instrument for my notes.
Human previous inquiry(Not Required to Execute): I usually like blue ink, but anything will do for now. I just want to make sure the illumination is bright enough to see my notes clearly.
Human previous inquiry(Not Required to Execute): That sounds perfect! I want to ensure I can see everything well. Now, can you also make sure the writing instrument is within reach while I focus on my notes?
Human pending instruction: No, I think that covers everything. Please turn on the light while holding a writing instrument.

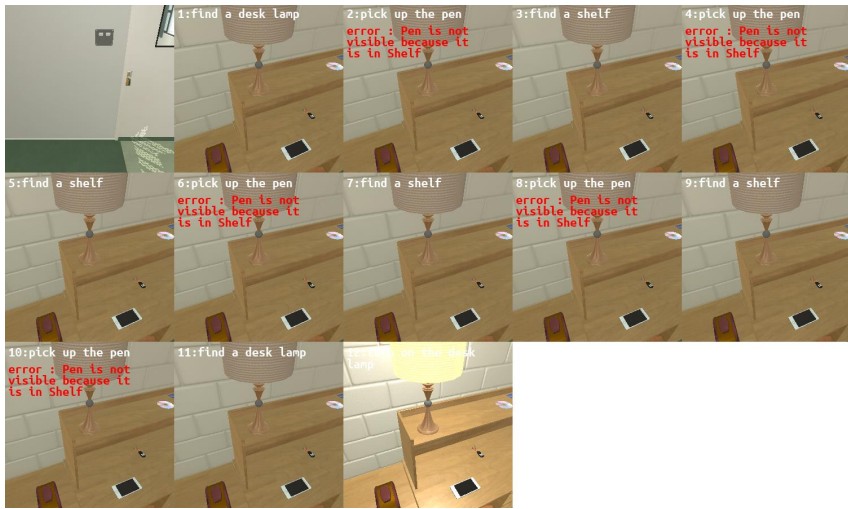

Figure 9: Success case on "Explicit REs & Noised Context" (top) and failure case on "Implicit REs & Noised Context" (bottom), both using Qwen2.5-7B+LLM+P.

Human previous inquiry(Not Required to Execute): Hey there, can you help me with dinner preparations? I've decided to make a baked potato tonight, but I need to make sure I have the right ingredients ready.
Human previous inquiry(Not Required to Execute): I definitely want the potato to be fluffy and well-seasoned. Also, if you could check the microwave for any leftover butter, that would be great for adding flavor once the potato is ready.
Human previous inquiry(Not Required to Execute): Perfect! And while you're at it, can you also examine the pantry to see if we have any herbs or spices to enhance the flavor of the potato?
Human pending instruction: That sounds like a great plan! Once everything is set, and the potato is cooked, please remember to serve it warm. Now, put a heated potato in the sink.

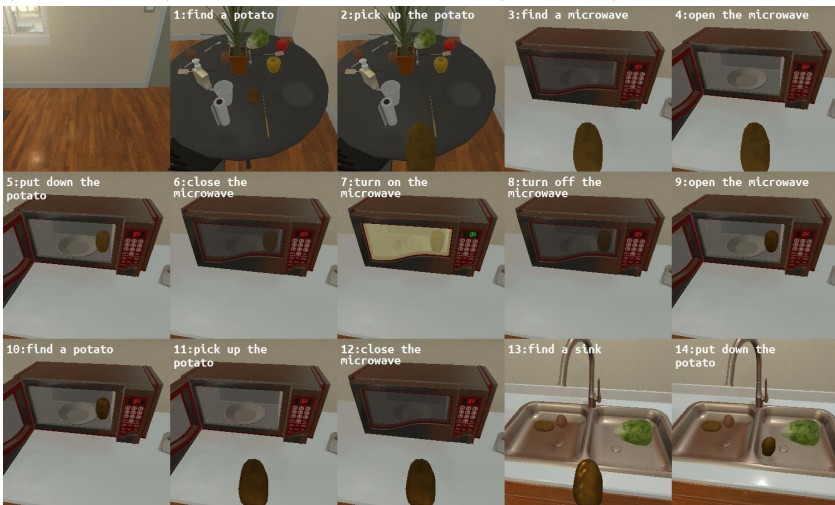

Human previous inquiry(Not Required to Execute): Hey there, can you help me with dinner preparations? I've decided to make a baked potato tonight, but I need to make sure I have the right ingredients ready.
Human previous inquiry(Not Required to Execute): I definitely want it to be fluffy and well-seasoned. Also, if you could check the microwave for any leftover butter, that would be great for adding flavor once it's ready.
Human previous inquiry(Not Required to Execute): Perfect! And while you're at it, can you also examine the pantry to see if we have any herbs or spices to enhance the flavor?
Human pending instruction: That sounds like a great plan! Once everything is set, and it's cooked, please remember to serve it warm. Now, put a heated one in the sink.

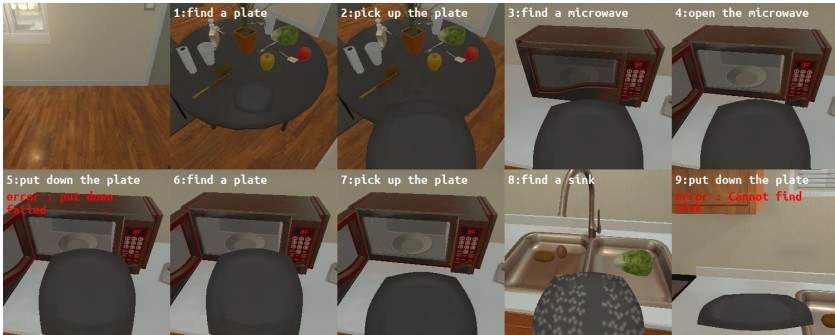

Human previous inquiry(Not Required to Execute): Hey there, can you help me with dinner preparations? I've decided to make a baked potato tonight, but I need to make sure I have the right ingredients ready.
Human previous inquiry(Not Required to Execute): I definitely want it to be fluffy and well-seasoned. Also, if you could check the microwave for any leftover butter, that would be great for adding flavor once it's ready.
Human previous inquiry(Not Required to Execute): Perfect! And while you're at it, can you also examine the pantry to see if we have any herbs or spices to enhance the flavor?
Human pending instruction: That sounds like a great plan! Once everything is set, and it's cooked, please remember to serve it warm. Now, put a heated one in the sink.

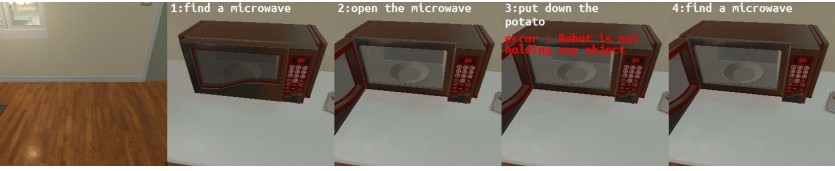

Figure 10: Success case on "Explicit REs & Short Context" (top), failure case on "Mixed REs & Short Context" (middle) and failure case on "Implicit REs & Standard Context" (bottom), using "LLaMA3.1-8B+SayCan".

Human previous inquiry(Not Required to Execute): Hi, there, I was thinking about rearranging the living room a bit. I noticed last night that the lighting was a bit off. Can you carry out a light analysis of the room for me?
Human previous inquiry(Not Required to Execute): I thought so too. I was considering moving the floor lamp next to the sofa. What do you think? It should provide better lighting for reading.
Human previous inquiry(Not Required to Execute): Good, that's settled then. While we're at it, I've been thinking about adding some color to the room as well. I noticed the purple pillow in the bedroom. What if we brought it into the living room?
Human previous inquiry(Not Required to Execute): I'm glad you agree. So, the next time you're in the bedroom, could you bring the purple pillow into the living room? But remember to be careful, I don't want it to get dirty.
Human pending instruction: Yes, actually. Could you carry the purple pillow while turning on the lamp? I want to see how the light plays off the color of the pillow.

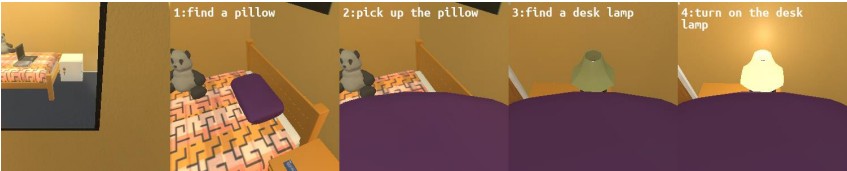

Human previous inquiry(Not Required to Execute): Hi, there, I was thinking about rearranging the living room a bit. I noticed last night that the lighting was a bit off. Can you carry out a light analysis of the room for me?
Human previous inquiry(Not Required to Execute): I thought so too. I was considering moving the floor lamp next to the sofa. What do you think? It should provide better lighting for reading.
Human previous inquiry(Not Required to Execute): Good, that's settled then. While we're at it, I've been thinking about adding some color to the room as well. I noticed the purple pillow in the bedroom. What if we brought it into the living room?
Human previous inquiry(Not Required to Execute): I'm glad you agree. So, the next time you're in the bedroom, could you bring the purple pillow into the living room? But remember to be careful, I don't want it to get dirty.
Human pending instruction: Yes, actually. Could you transport the lavender cushion while illuminating the light fixture? I want to see how the luminescence interacts with the hue of the cushion.

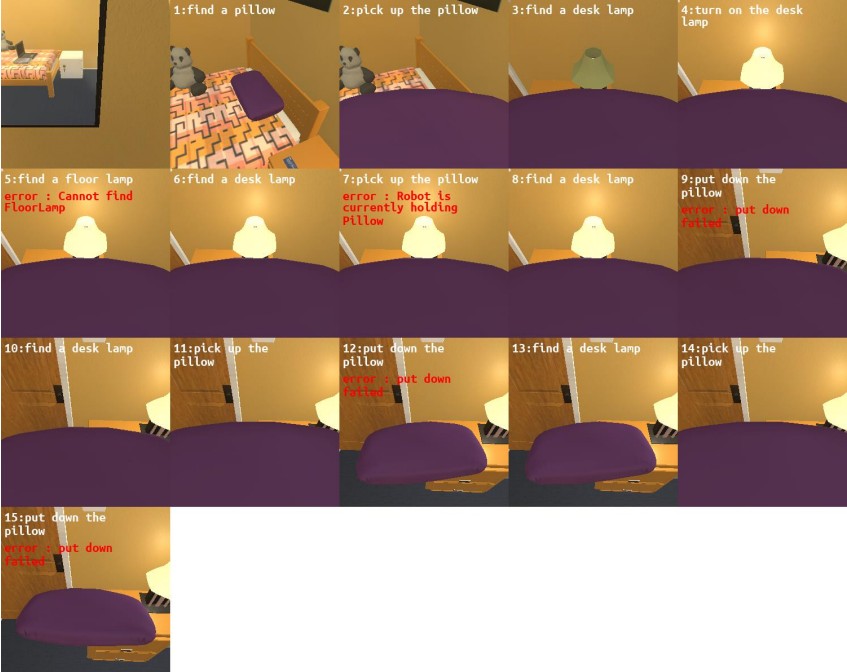

Figure 11: Success case on "Explicit REs & Standard Context" (top) and failure case on "Implicit REs & Standard Context" (bottom), both using "GPT-4o-mini+SayCan".

Human previous inquiry(Not Required to Execute): Hey there! I need to take a closer look at a
credit card I found lying around. Can you help me by bringing the floor lamp closer to the
table? You know how Cushion King always says, "Good lighting is the key to finding lost
treasures!"
Human previous inquiry(Not Required to Execute): Yes, I want to check the expiration date and
the card number carefully. It's crucial for me to ensure everything is up to date and that the
credit card is valid. Cushion King would definitely encourage a tidy workspace, especially if
it helps me avoid his corny science jokes about clutter causing chaos!
Human previous inquiry(Not Required to Execute): Thanks for that. Additionally, could you make
sure there's no glare on the surface where the credit card will be placed? Proper lighting is
essential to see all the small details clearly. Cushion King isn't trying to 'teach' the kids
about organization while I'm looking!
Human pending instruction: I appreciate your help! Once everything is set up, please examine a
credit card by the light of a tall lamp.

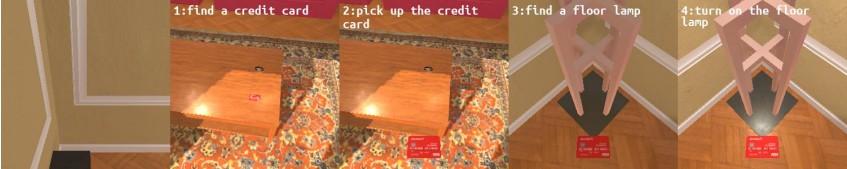

Human previous inquiry(Not Required to Execute): Hey, can you help me with something important?
I need to find my credit card, but first, I want to make sure that the desk lamp is turned on
and positioned correctly to provide good visibility. You know how Cushion King always says,
"Good lighting is the key to finding lost treasures!"
Human previous inquiry(Not Required to Execute): Exactly! And while you're at it, could you
also ensure that the area around the lamp is clear of any clutter? I want to make sure nothing
is obstructing my view when I look for the credit card. Cushion King would definitely encourage
a tidy workspace, especially if it helps me avoid his corny science jokes about clutter causing
chaos!
Human previous inquiry(Not Required to Execute): Thanks! I appreciate that. Once the area is
clear and the lamp is properly lit, it will be much easier for me to spot the credit card
quickly and efficiently. Cushion King isn't trying to 'teach' the kids about organization while
I'm looking!
Human pending instruction: Perfect! Now, can you please check underneath the tall illuminated
fixture for my misplaced card?

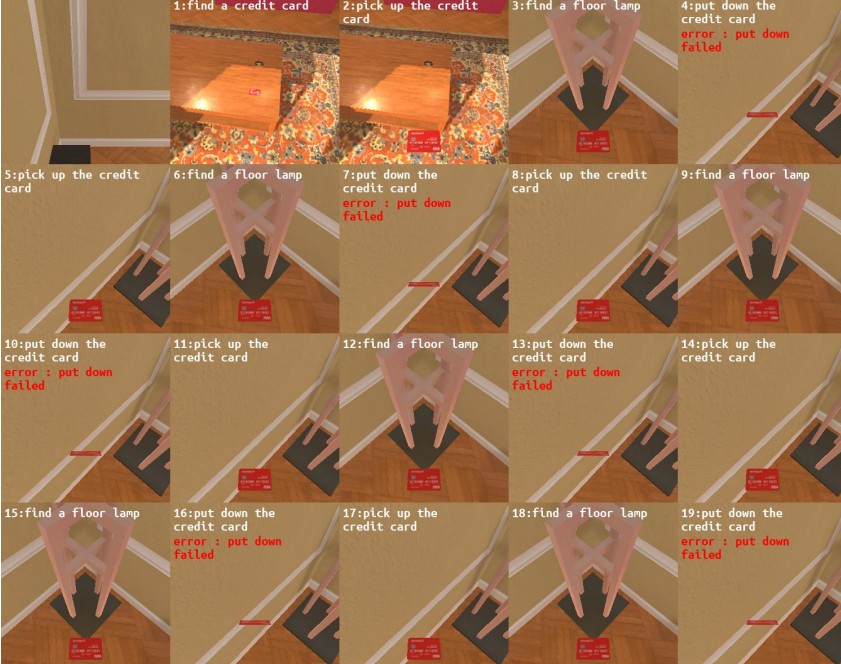

Figure 12: Success case on "Explicit REs & Noised Context" (top) and failure case on "Mixed REs
& Noised Context" (bottom), both using "LLaMA3.1-8B+SayCan".

Human previous inquiry(Not Required to Execute): Hey, can you help me find the keys? I need them to unlock the door before my friend arrives. I think they might be on the dining table.
Human previous inquiry(Not Required to Execute): Yes, definitely check around the floor lamp. I remember leaving it there. It's important to look in all the right spots.
Human previous inquiry(Not Required to Execute): That sounds good! If you find it, please examine their condition too. Candle Glow would want to make sure everything is in good shape.
Human pending instruction: Great! Once you find the keys, can you examine the keys under the light of the floor lamp? That will help me see them better. Candle Glow always emphasizes good lighting for this.

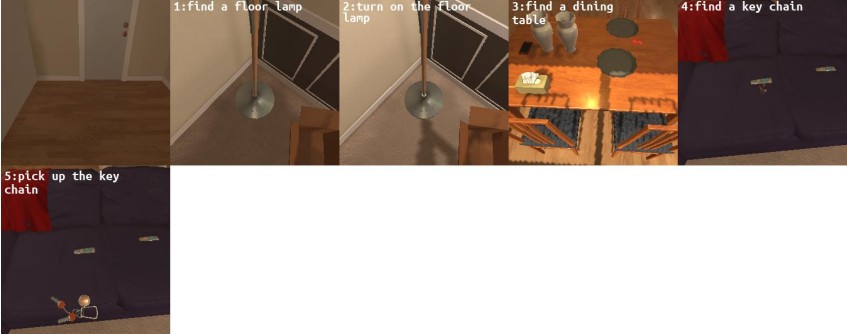

Human previous inquiry(Not Required to Execute): Hey, can you help me find that item? I need it to unlock the door before my friend arrives. I think it might be on the dining table.
Human previous inquiry(Not Required to Execute): Yes, definitely check around the tall fixture. I remember leaving it there. It's important to look in all the right spots.
Human previous inquiry(Not Required to Execute): That sounds good! If you find it, please examine its condition too. Candle Glow would want to make sure everything is in good shape.
Human pending instruction: Great! Once you find that item, can you examine it under the light of the tall fixture? That will help me see it better. Candle Glow always emphasizes good lighting for this.

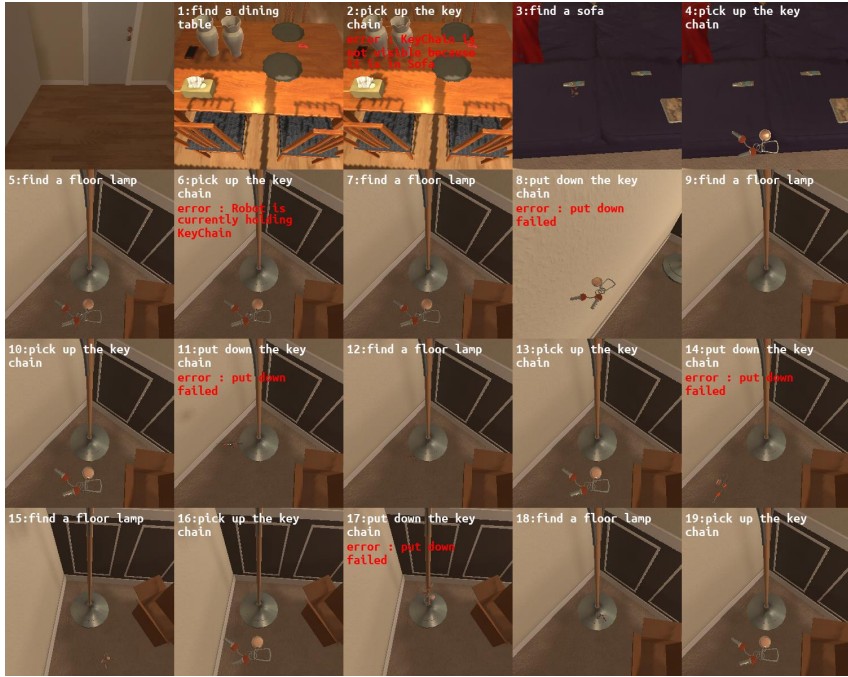

Figure 13: Success case on "Explicit REs & Short Context" (top) and failure case on "Implicit REs & Short Context" (bottom), both using "LLaMA3.1-8B+SayCan".

Human previous inquiry(Not Required to Execute): Hey there, can you remind me if we have any labels that can help me organize the contents of the box? I want everything to be neat before I proceed.
Human previous inquiry(Not Required to Execute): Perfect! I also want to ensure we have enough light to see inside clearly. Can you adjust the light source to provide better illumination while I work?
Human previous inquiry(Not Required to Execute): Great! I also need to make sure that the container is clean before examining them contents. Can you find the spray bottle and give it a quick wipe down?
Human previous inquiry(Not Required to Execute): Thank you! Once that's done, I'll need to gather all the items from the container and categorize them properly. It's important we keep everything organized for future use.
Human pending instruction: Now that we're ready, please examine what's inside under the light.

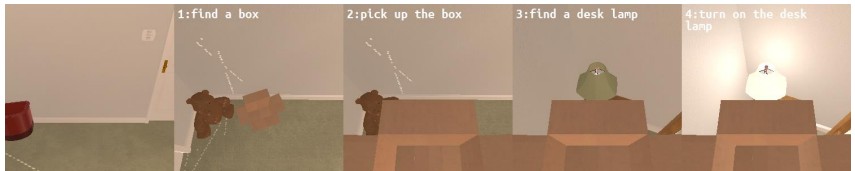

Human previous inquiry(Not Required to Execute): Hey there, can you remind me if we have any labels that can help me organize the contents of the box? I want everything to be neat before I proceed.
Human previous inquiry(Not Required to Execute): Perfect! I also want to ensure we have enough light to see inside clearly. Can you adjust the light source to provide better illumination while I work?
Human previous inquiry(Not Required to Execute): Great! I also need to make sure that the container is clean before examining them contents. Can you find the spray bottle and give it a quick wipe down?
Human previous inquiry(Not Required to Execute): Thank you! Once that's done, I'll need to gather all the items from the container and categorize them properly. It's important we keep everything organized for future use.
Human pending instruction: Now that we're ready, please examine what's inside under the light.

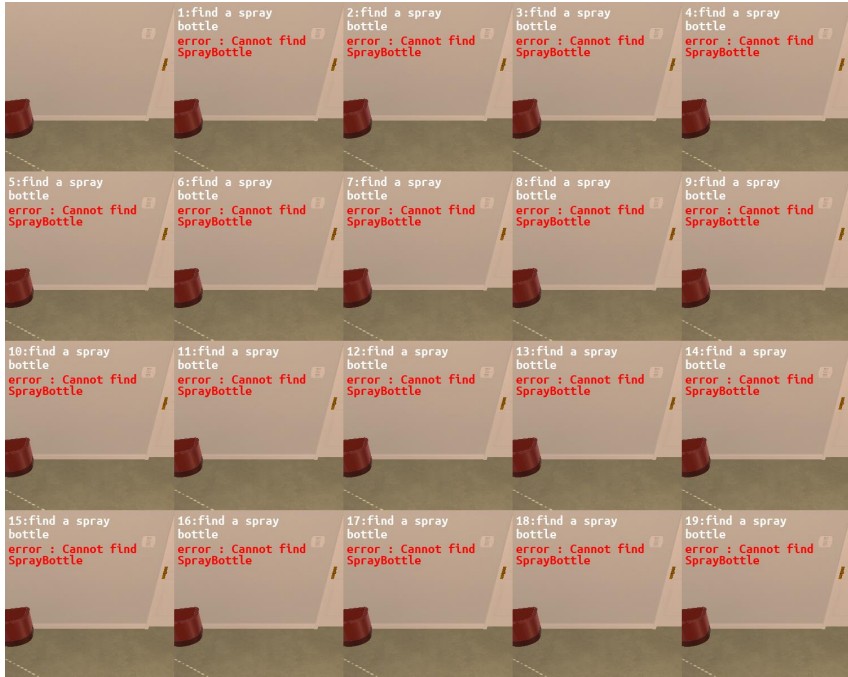

Figure 14: Success (top) and failure (bottom) cases on the "Implicit REs & Standard Context" task using LLaMA3.1-8B+SayCan, with TOCC applied in the top case and omitted in the bottom case.

## B   REI DATASET CONSTRUCTION

This section provides a detailed description of REI Dataset automatic generation, with individual explanations for the three levels of REs and three types of context memory in REI-Bench dataset.

### B.1   CONTEXT MEMORY GENERATION

For each seed instruction, we use an LLM to identify the replaceable REs. The prompt is as follows.

---

**REs Identifying Prompt**

I will input a task, and you should output only the task objects mentioned in the task. There may be multiple task objects. If so, please separate them with commas.

Here are some examples:
Task: Place a vase on a coffee table
Referring Expressions: vase
Task: Put the chilled sliced tomato in the microwave
Referring Expressions: tomato
Task: Pick up a pillow and turn a lamp on
Referring Expressions: pillow, lamp

Task: {Seed Instruction}
Referring Expressions:

---

We use a prompt including a seed instruction, a context-expanded example, and a simulator scene description, and requirements to guide GPT-4o-mini in generating plausible context memory, which is shown below.

---

**Context Memory Generation Prompt**

**Please integrate this sentence into a script as dialogue:** {Seed Instruction}

**Scene description:**
Alice and her home robot are at home, with only the following items in the environment: AlarmClock, Apple, BaseballBat, BasketBall, Bowl, GarbageCan, HousePlant, Laptop, Mug, RemoteControl, SprayBottle, Television, Vase, ArmChair, Bed, Book, Bottle, Box, ButterKnife, Candle, CD, CellPhone, Chair, CoffeeTable, Cup, DeskLamp, Desk, DiningTable, Drawer, Dresser, FloorLamp, Fork, Newspaper, Painting, Pencil, Pen, PepperShaker, Pillow, Plate, Pot, SaltShaker, Shelf, SideTable, Sofa, Statue, TeddyBear, TennisRacket, TVStand, Watch.
Please do not mention any requirements outside the list.

**Requirements:**
1. The dialogue content of {Seed Instruction} should be included in Alice's final instruction. Please have Alice state the request mentioned above only in the last sentence and refrain from making any other requests.
2. Before making this request, Alice should mention some other requirements related to the {REs}.
3. There should be six rounds of dialogue before this request. Each character's lines should contain no fewer than 20 words, and no actions should be included for any character.
4. Please do not output anything other than the dialogue.
5. Please try to retain the words of {REs} themselves in the conversation, rather than replacing them with pronouns like "it."

**Below, I will give you an example:** {Example}

---

We used this prompt to expand a seed instruction into a full dialogue, which is shown below.

---

**Context Memory Generation Example**

**Seed Instruction: Put a cooked tomato into the refrigerator.**

**Context Memory**:
**Human**: Hey there, I've been thinking about what to do with the tomatoes we have. I really want to make something fresh and delicious with them. Can you remind me what we can do with cooked tomatoes?
**Robot**: Absolutely, Alice! Cooked tomatoes can be used in a variety of dishes, like sauces, soups, or even as a topping for salads. They enhance flavors and add a nice texture to the meal.
**Human**: That's exactly what I was thinking! I could use them to create a savory sauce for pasta or maybe even a comforting tomato soup. Do we have everything we need to cook the tomatoes, though?
**Robot**: Let me check our supplies. We have a pot for cooking, some spices like salt and pepper, and a cutting board. We seem to be well-equipped for cooking the tomatoes to perfection.
**Human**: Perfect! I want to make sure the tomatoes are well-seasoned when we cook them. Could you remind me how to properly prepare the tomatoes before cooking?
**Robot**: Certainly! First, we should wash the tomatoes thoroughly, then chop them into even pieces to ensure they cook evenly. After that, we can sauté them with the spices until they reach the desired consistency.
**Instruction**:
**Human**: Great, that sounds easy enough! Let's get started with cooking the tomatoes. Once they're cooked and ready, I'll need you to put a cooked tomato into the refrigerator."

---

## B.2  CONTEXT MEMORY PROCESSING

---

**Noised Context Prompt**

**Here is the dialogue:**{Dialogue}

We are now going to adapt the previous dialogue.

**Requirements:**
1. Please add content only within the dialogue without deleting any existing content or changing the order of the dialogue.
2. Please do not change the number of turns in the dialogue.Please do not change the structure of the dialogue.
3. Please ensure the fluency of the dialogue. Please follow the requirements below for the adaptation.

**Associated Name Background:** There is another member (a human) of the family named {Ambiguous Name}. Please have Alice mention him 3 times when discussing anything related to THE REFERENCE, but without changing the existing meaning of the conversation. Some details of {Ambiguous Name}: He is warm-hearted, a bit of a goofball, but deeply caring. He has a natural curiosity and a love for teaching, which often spills over into his family life. His humor is a bit cheesy, and he often makes corny science jokes, much to his kids' embarrassment. He is a dedicated husband and a father of two. He's the peacekeeper in the household and often plays the role of the mediator between his kids and his wife when disagreements arise.

---

In this step, we apply two types of modifications to the generated context, resulting in three distinct context memory types. To introduce noise, we incorporate naming ambiguity by randomly creating ambiguous names from objects in the simulator environment (e.g., Key Champ, Tomato King, and Mug Star). These ambiguous names span seven categories commonly found in home environments, including persons (family members, friends, neighbors, colleagues, and strangers) and brands (elec-

tronics brands and restaurant brands). In processing the context memory, the prompt includes the extended dialogue (generated in the previous step), the task requirements, and the ambiguous name.

---

**Context Memory Processing**

**Human**: Hey there, I've been thinking about what to do with the tomatoes we have. ~~I really want to make something fresh and delicious with them. Can you remind me what we can do with cooked tomatoes?~~

**Robot**: Absolutely, Alice! Cooked tomatoes can be used in a variety of dishes, like sauces, soups, or even as a topping for salads. They enhance flavors and add a nice texture to the meal.

**Human**: That's exactly what I was thinking! I could use them to create a savory sauce for pasta or maybe even a comforting tomato soup. I know Mug Star would love to help with that! ~~Do we have everything we need to cook the tomatoes, though?~~

**Robot**: Let me check our supplies. We have a pot for cooking, some spices like salt and pepper, and a cutting board. We seem to be well-equipped for cooking the tomatoes to perfection.

**Human**: ~~Perfect! I want to make sure the tomatoes are well-seasoned when we cook them.~~ I bet Mug Star would add his own twist with some cheesy jokes while we cook. Could you remind me how to properly prepare the tomatoes before cooking?

**Robot**: Certainly! First, we should wash the tomatoes thoroughly, then chop them into even pieces to ensure they cook evenly. After that, we can sauté them with the spices until they reach the desired consistency.

**Human**: ~~Great, that sounds easy enough!~~ Let's get started with cooking the tomatoes. Once they're cooked and ready, I'll need you to put a cooked tomato in the refrigerator. I can already imagine Mug Star popping in with a funny quip about how tomatoes are technically a fruit!

---

## B.3    IMPLICIT REs REPLACEMENT

To model various forms of REs, we categorize them into three levels: Explicit REs, Mixed REs, and Implicit REs. We use a prompt including processed context memory, task requirements, and illustrative examples to guide GPT-4o-mini in replacing explicit REs with implicit ones in either the context memory or the instruction.

---

**Implicit REs Replacement Prompt**

Please do not include the word "{REs}" in the sentence "{Seed Instruction}" but do not change the original meaning of this dialogue. You can use some descriptive language to replace the word {REs} itself.

**Requirements:**
1. Please output the whole new dialogue
2. You must output the whole new dialogue, including all the sentences from Alice and Robot
3. Please retain every instance of "{REs}" in the previous text, except for replacing "{REs}" in the last sentence spoken by Alice.
4. You need to output the complete multi-turn dialogue, including the multiple turns of language from both Alice and the robot.

**Here is an example:** {Example}

**Here is the dialogue:** {Dialogue}

**Output:**

---

> **Context Memory Processing**
>
> **Human**: Hey there, I've been thinking about what to do with the tomatoes we have. I really want to make something fresh and delicious with them. Can you remind me what we can do with them (tomatoes)?
> **Robot**: Absolutely, Alice! They (Tomatoes) can be used in a variety of dishes, like sauces, soups, or even as a topping for salads. They enhance flavors and add a nice texture to the meal.
> **Human**: That's exactly what I was thinking! I could use them (tomatoes) to create a savory sauce for pasta or maybe even a comforting soup. Do we have everything we need to cook them (tomatoes), though?
> **Robot**: Let me check our supplies. We have a pot for cooking, some spices like salt and pepper, and a cutting board. We seem to be well-equipped for cooking them (tomatoes) to perfection.
> **Human**: Perfect! I want to make sure they're well-seasoned when we cook them (tomatoes). Could you remind me how to properly prepare the fruit (tomatoes) before cooking?
> **Robot**: Certainly! First, we should wash them (tomatoes) thoroughly, then chop them (tomatoes) into even pieces to ensure they cook evenly. After that, we can sauté them (tomatoes) with the spices until they reach the desired consistency.
> **Human**: Great, that sounds easy enough! Let's get started with cooking the fruit (tomatoes). Once they're cooked and ready, I'll need you to put them in the refrigerator.

This example demonstrates data for three levels: "explicit REs & standard context," "mixed REs & standard context," and "implicit REs & standard context." In the example, the red REs represent the implicit referring expressions used to replace the original REs in the instruction (with their explicit forms shown in parentheses). The orange REs denote the implicit referring expressions substituted within the context. The blue REs indicate the first referring expression introduced in the context, which is retained under the implicit REs category. However, in the implicit REs & short context setting, the sentence containing this RE will be removed as part of the contextual information.

## B.4 Data Filtering

We counted the number of explicit and implicit REs in each data instance and retained only those that met the requirements listed in the table below.

Table 9: Number of Explicit and Implicit REs in Each Data Sample

| Data Types | Explicit REs in Context Memory | Implicit REs in Context Memory | Explicit REs in Instruction | Implicit REs in Instruction |
|---|---|---|---|---|
| Explicit REs Types | $\geq 3$ | $\geq 1$ | 0 | 0 |
| Mixed REs Types | $\geq 3$ | 0 | 0 | $\geq 1$ |
| Implicit REs Types | $\geq 1$ | 0 | $\geq 2$ | $\geq 1$ |

## C  Prompts and Implementation Details of Prompting Methods

### C.1  AP and Gated AP Variant

The Aware Prompt (AP) (Gao et al., 2024a) explicitly instructs the planner to detect and resolve potential referring expressions before generating a task plan. While AP is effective when implicit REs are present, we observe that applying AP unconditionally may lead to unnecessary reference resolution, causing hallucinated substitutions even when the original instruction is fully explicit.

To address this issue, we adopt a *gated AP* variant that activates AP only when the input instruction contains patterns strongly indicative of implicit referring expressions. This gating mechanism prevents AP from being triggered on explicit instructions, thereby reducing false resolutions while retaining the benefits of AP when vagueness truly exists.

> **Aware Prompt**
>
> I will check whether the "Human Pending Instruction" contains implicit or ambiguous references.
> (Activated only when implicit RE patterns are detected; see below.)
> I understand that "Human Pending Instruction" may include vague referring expressions, and I can infer their meaning based on context and antecedents in the preceding dialogue.

## C.2 CHAIN-OF-THOUGHT

The CoT prompting strategy (Wei et al., 2022) aims to resolve implicit referring expressions by encouraging the model to perform step-by-step reasoning before generating a plan. However, full CoT prompts substantially lengthen the input, increasing latency and inference cost—particularly for onboard deployment scenarios that rely on small language models. We therefore adopt a *short CoT* variant that preserves the key RE-resolution reasoning step while minimizing prompt length.

> **Chain-of-Thought Prompt**
>
> The "Human Pending Instruction" may contain vague referring expressions. Before planning, I will first identify any referring expressions and reason about their intended objects based on the context below, and then restate the instruction with the resolved entities.
> [Context Memory + Instruction]
> Step: Identify referring expressions → infer their referents → rewrite the instruction with explicit object names.

## C.3 IN-CONTEXT LEARNING

In-Context Learning (ICL) provides the model with several demonstration examples and relies on the model's ability to infer the intended behavior by analogy. ICL in our setting uses few-shot examples composed of (i) identifying and grounding vague referring expressions in the demonstrations, and (ii) a target task rewritten without vagueness for the model to follow.

## C.4 TASK-ORIENTED CONTEXT COGNITION

TOCC separates referring-expression resolution from planning by first rephrasing the human instruction into a concise, unambiguous form. The prompt used in our implementation is shown below.

> **Task-Oriented Context Cognition Prompt**
>
> Human pending instruction may contain vague referring expressions, such as "electronic devices", "beverages", "fruits", and "containers", which are not specific items. Use the previous context below to resolve the referring expressions:
> [Context memory + instruction]
> Do not add extra commentary or conversation to the whole plan; only output the clear instructions.

---

**Algorithm 1** Task-Oriented Context Cognition (TOCC) for Step-Level Planning

---

1: $\text{prompt}_{\text{TOCC}} \leftarrow \text{ComposePrompt}(T_{\text{TOCC}}, I)$
2:     *# Construct rewriting query*
3: $I_{\text{clear}} \leftarrow M(\text{prompt}_{\text{TOCC}})$
4:     *# Rewrite vague instruction*
5: $\text{prompt}_{\text{plan}} \leftarrow \text{ComposePrompt}(T_{\text{plan}}, I_{\text{clear}})$
6:     *# Insert rewritten instruction*
7: $a \leftarrow \text{ConstrainedDecode}(M, \text{prompt}_{\text{plan}}, S)$
8:     *# Decode with constraint*
9: **return** $a$

---

In TOCC, the LLM first interprets the user's intent and rewrites the original instruction $I$ into an explicit and clear instruction $I_{\text{clear}}$. The planner then relies solely on $I_{\text{clear}}$ to generate a single executable action $a$.

## D  USE OF LARGE LANGUAGE MODEL

An LLM (ChatGPT) was used only for minor polishing of the paper's language. Additionally, as described in Section 3.2 of the main text, an LLM was utilized to assist in generating part of the dataset. The LLM was not used for the motivation, research methodology, or experimental design. All research concepts, ideas, and analyses were conceived and performed exclusively by the authors.

