# OpenReview forum: "REI-Bench: Can Embodied Agents Understand Vague Human Instructions in Task Planning?"
_ICLR.cc/2026/Conference — ICLR 2026 Poster_

### Official Review · Reviewer_YMUn · 2025-10-16

**Soundness:** 3
**Presentation:** 3
**Contribution:** 2
**Rating:** 6
**Confidence:** 3

**Summary:**

This paper is the first to identify and study the challenge of LLM-based robot task planning when vague human instructions arise from referring expressions (REs). The authors propose a novel benchmark, REI-Bench, to evaluate the impact of such vagueness and demonstrate that implicit REs can significantly degrade the performance of planners. To address this, they introduce TOCC, a simple yet effective approach to mitigate the effects of vagueness, achieving superior performance compared to existing baselines.

**Strengths:**

1. The paper introduces a noval benchmark to simulate vagueness in human instructions which allows for a standardized evaluation of how vagueness impacts robot task planning.
2. The problem of interpreting vague instructions is crucial for making robots more accessible. This work has practical implications in real-world environments.
3. This paper conduct extensive experiments on several baseslines and demonstrates the proposed TOCC outperforms them.

**Weaknesses:**

1. The paper focuses primarily on a small subset of REs, specifically coreferential vagueness. Can TOCC be extended to address more descriptive phrases such as "the heavy thing"?
2. The evaluation is limited to language models with relatively small model sizes. Testing on more powerful models as a point of comparison would help contextualize the results better.

**Questions:**

1. Can the framework be extended to address other types of vagueness beyond coreferential vagueness? How easily can it be extended, and what would be the performance implications?
2. Will modern, larger models be able to resolve the vagueness challenge more effectively?

---

> ### Author Response · Authors · 2025-11-20
> **Response to Reviewer YMUn (1/2)**
>
> We sincerely thank Reviewer **YMUn** for the insightful and constructive comments. We are truly encouraged that the reviewer recognizes **“this work has practical implications in real-world environments”.** In the following, we address all concerns raised by the reviewer and provide further analyses, statistics, and clarifications to strengthen the manuscript. We hope the responses can address the concerns. In addition, we have submitted a revised manuscript with an appendix where we mark all the suggested tables, figures, and analytics in magenta color.
>
> **W1 & Q1: Can the framework be extended to address other types of vagueness beyond coreferential vagueness? How easily can it be extended? Can TOCC be extended to address more descriptive phrases, such as "the heavy thing", and what would be the performance implications?**
>
> **Answer:**
> TOCC can resolve more descriptive phrases. In REI-Bench, we model two types of implicit REs, one of which is descriptive expressions, for example,  “the light source” (referring to a lamp), “the container” (referring to a box), or “the sweet fruit” (referring to an apple). TOCC can correctly rewrite these expressions and resolve the corresponding ambiguity. Simplified examples of such clarifications are shown in Tab. 1.
>
> Furthermore, TOCC can be extended to other forms of linguistic vagueness with only minor modifications and is expected in principle to yield performance gains comparable to those observed for REs. The reason is that TOCC targets a common weakness shared by many planners: prioritizing rapid task execution over the cognitive effort required to understand the instruction correctly. This issue persists regardless of the specific type of vagueness involved. TOCC mitigates it by explicitly enforcing an interpretation and clarification step, ensuring that the planner identifies the intended meaning before generating executable actions. For example, older adults may frequently repeat phrases or interleave their actual intent with irrelevant speech, thereby introducing ambiguity. In such cases, TOCC can isolate the genuinely actionable instruction rather than letting the planner be distracted by unrelated content.
>
> **Table 1. Simplified examples of TOCC clarifying descriptive expressions**
>
> | Context (Simplified)                                      | Vague Instruction                                        | Clear Instruction (TOCC)          |
> | --------------------------------------------------------- | -------------------------------------------------------- | --------------------------------- |
> | I could really use an **apple** salad right now.          | I wish you could place **the sweet fruit** on the table. | Put **the apple** on the table.   |
> | I might find a **box** and pack some snacks for the trip. | Could you put **the storage thing** on the chair?        | Put **the box** on the chair.     |
> | I want to **brighten up** the room a bit.                 | Please check the card with **the  light source**.        | Check the card with **the lamp**. |
> | I need something to **stir the soup**.                    | Can you put **the long metal tool** into the bowl?       | Put **the spoon** into the bowl   |
>
>
> Moreover, we plan to extend the framework in REI-Bench 2.0 to additional vagueness types and evaluate it on more realistic human data, which we are currently preparing under Institutional Review Board (IRB) review. A comprehensive comparison of TOCC’s performance across these new settings will be included at that stage.

---

> ### Author Response · Authors · 2025-11-20
> **Response to Reviewer YMUn (2/2)**
>
> **W2 & Q2: Will modern, larger models be able to resolve the vagueness challenge more effectively?**
>
> **Answer:**
> Yes, larger models can address vagueness more effectively. However, we primarily consider small open-source models for two reasons. First, in real-world robotic settings, models must operate under strict computational constraints, such as household robots, making small-scale models more practical for deployment. Second, on-robot deployment typically requires further adaptation, such as fine-tuning or reinforcement learning, which in turn necessitates the use of open-source models. These considerations are presented in the Introduction and Section 4.1 of the manuscript.
>
> Nevertheless, to provide a more complete benchmark comparison, we include additional experiments. As shown in Tab. 1, the success rate of “GPT-4o + SayCan” improves by a mean of 32.0% over “gpt-4o-mini” but remains 28.0% below the human baseline on average. This suggests that larger models cope with vague language more effectively, while still leaving considerable room for improvement compared to human performance. Tab. 1 has been incorporated into Fig. 4 of the revised manuscript.
>
> **Table 1: Success rates (%) of GPT-4o and GPT-4o-mini using the SayCan framework, compared with the human baseline. Results under three types of contexts for three types of REs are reported.**
>
>
> |                      |                  | GPT-4o-mini + SayCan |                  |                  | GPT-4o + SayCan |                  |                  | Human Baseline |                  |
> | -------------------- | :--------------: | :------------------: | :--------------: | :--------------: | :-------------: | :--------------: | :--------------: | :------------: | :--------------: |
> |                      | **Explicit REs** |    **Mixed REs**     | **Implicit REs** | **Explicit REs** |  **Mixed REs**  | **Implicit REs** | **Explicit REs** | **Mixed REs**  | **Implicit REs** |
> | **Standard Context** |      45.00       |        25.90         |      24.30       |      73.50       |      61.60      |      52.60       |       97.0       |      91.0      |       86.0       |
> | **Noised Context**   |      43.10       |        26.20         |      21.30       |      72.90       |      61.20      |      52.40       |       95.0       |      92.0      |       85.0       |
> | **Short Context**    |      44.40       |        23.30         |      21.30       |      73.10       |      61.80      |      52.90       |       98.0       |      91.0      |       79.0       |

---

> > ### Comment · Reviewer_YMUn · 2025-11-24
> > **Thanks for your response**
> >
> > Thank you for the detailed response. The authors have addressed most of my concerns, and I will keep my current rating.

---

> > > ### Author Response · Authors · 2025-11-24
> > > **Appreciation for the constructive comments**
> > >
> > > We sincerely appreciate your helpful suggestions! Thanks for your time and effort!

---

> ### Author Response · Authors · 2025-11-24
> **Looking forward to your reply**
>
> Dear Reviewer YMUn,
>
> As the rebuttal deadline is approaching, could you please have a look at our response? We are looking forward to your reply!
>
> Feel free to let us know if you have any other concerns. Thanks for your time and effort!
>
> Best Regards,
>
> Authors of Submission 3905

---

### Official Review · Reviewer_huym · 2025-10-25

**Soundness:** 2
**Presentation:** 2
**Contribution:** 1
**Rating:** 2
**Confidence:** 4

**Summary:**

The paper shows that  referring expressions (REs),a source of coreferential vagueness would significantly degrade LLM-based embodied task planning. It introduces **REI-Bench**, a benchmark that systematically varies RE vagueness and dialogue context, and reports up to a 36.9% absolute drop in success for off-the-shelf planners (SayCan, LLM+P). The authors propose Task-Oriented Context Cognition (TOCC), a lightweight preprocessing step that resolves REs before planning, reducing object-omission errors across nine difficulty levels. The work surfaces an overlooked failure mode in HRI and offers both a rigorous evaluation protocol and a practical, immediately deployable mitigation.

**Strengths:**

1. Clear problem framing motivation. The paper isolates  coreferential vagueness from referring expressions (REs) as a specific, under-explored failure mode in embodied task planning, grounding the setup in linguistic theory (RE vs. DE, bridge inference).

2. Salient empirical findings. The work documents  consistent, sizable drops in task success under implicit REs and multi-turn dialogue, and includes a human reference to highlight the performance gap in natural conversational settings.

**Weaknesses:**

1. I have substantial concerns about how the benchmark is constructed, this is the main issue, as the dataset’s design directly affects the trustworthiness of the empirical results.
  * Sampling/selection bias: Filtering out Pick Two & Place and seeding only from tasks successfully executed by LLaMA3.1-8B+SayCan simplifies the benchmark and risks biasing task distribution; no unfiltered vs. filtered comparison is reported to rule out selection effects.
  * The Ambiguous-Name perturbation (brand/person name collisions) is too narrow and misses common home scenarios (homophones,, family/pet/room names, and visual look-alikes like pot/pan/bowl), limiting generalization beyond the injected pattern. More ecologically valid generators of referential ambiguity would strengthen the dataset’s realism and external validity.
  * Although the paper argues vagueness is more common for non-experts (elderly/children), the dataset does not model age-specific linguistic phenomena nor provide any evidence to test this claim.

2. “First coreferential-vagueness planning benchmark” claim lacks quantitative positioning. The paper does not provide a systematic, quantitative comparison against prior ambiguity/RE/coreference datasets across robotics, NLP, and HRI, nor a detailed dataset-level contrast table.

3. TOCC is under-specified at the algorithmic level. There is no precise interface or flow diagram: inputs/outputs, whether TOCC emits a disambiguated “clear instruction”, and whether it returns executable targets

4. Confounds in empirical comparisons. Although §3.3 claims CoT/ICL substantially lengthen prompts, AP/CoT/ICL are compared to TOCC without length comparisions, normalizing token cost/latency. I also concers about the fairness for baselines, which are under-specified.
Baseline improvements could be stronger with gated AP, short/segmented CoT, or ICL distillation; these are not reported, so TOCC’s margin may be overstated.

**Questions:**

1. Please address the concerns summarized under Weakness 1.
2. Can you provide a quantitative comparison table vs. prior ambiguity/RE/coreference datasets across Robotics/NLP/HRI ?A clear table and analysis would show REI-Bench fills a unique, previously uncovered evaluation axis (RE-driven vagueness in planning).
3. Provide an explicit and detailed description of TOCC, such as pseudocode or  data-flow diagram. The pipeline should be precise enough to reimplement.
4. Compared with stronger  variants of baselines with length comparisions, normalizing token cost/latency.

---

> ### Author Response · Authors · 2025-11-20
> **Response to Reviewer huym (1/5)**
>
> We sincerely thank Reviewer **huym** for the insightful, detailed, and valuable comments. Although many issues and critical observations were raised, these technically grounded questions have significantly helped us refine and improve the quality of the manuscript. In particular, the reviewer’s suggestions guided us to **strengthen the rigor of our experiments, avoid potential overclaims, and enhance the fairness and completeness of our methodological comparisons.**
>
> In the following, we address all concerns point-by-point and provide additional analyses, statistics, and clarifications to further reinforce the manuscript. We hope that our responses satisfactorily resolve the reviewer’s concerns. In addition, we have submitted a revised manuscript with an appendix, in which all newly added tables, figures, and analyses are highlighted in magenta for clarity.
>
>
> **Q1-1 (W1-1): Why do we exclude “Pick Two & Place” and filter the task that can’t be successfully finished by LLaMA3.1-8B + SayCan? Does this filtering introduce a selection bias?**
>
> **Answer:**
> Our motivation is to investigate how implicit REs affect the planner's performance. To isolate this effect, we adopt a controlled-variable approach: we select tasks where models can reliably complete the original clear instructions, and then introduce RE instructions to measure the resulting performance drop. This setup enables a focused and systematic analysis of referential vagueness.
>
> To address potential sampling or selection effects, we apply *stratified sampling* before the experiment to preserve ALFRED’s original task-type distribution. As shown in Tab. 1, the sampled dataset closely matches the original task proportions, suggesting that any sampling-induced distribution shift is minimal. Tab. 1 has been added to Appendix A.2, and the corresponding revisions have been incorporated into Section 4.1 Experiment Setup in the manuscript.
>
> **Table 1: Task-type distribution and average subtask steps in the 1,000-task sampled subset. The column “Original Proportion” corresponds to the original ALFRED dataset distribution,  and our stratified sampling preserves this proportion.**
>
> | Task Type           | Original Proportion (%) | Sampled Count |
> | ------------------- | ----------------------- | ------------- |
> | Cool & Place        | 16.8%                   | 168           |
> | Heat & Place        | 16.8%                   | 168           |
> | Clean & Place       | 16.2%                   | 162           |
> | Examine in Light    | 13.3%                   | 133           |
> | Stack & Place       | 18.4%                   | 184           |
> | Pick & Place        | 18.5%                   | 185           |
> | Pick Two (Excluded) | ---                     | 0             |
> | Total          | 100%               | 1000      |

---

> ### Author Response · Authors · 2025-11-20
> **Response to Reviewer huym (2/5)**
>
> **Q1-2 (W1-2): Is the Ambiguous-Name perturbation too narrow in scope?**
>
> **Answer:**
> We agree that Ambiguous-Name perturbation represents only one type of linguistic ambiguity. However, our goal is not to enumerate all possible perturbation phenomena, but to systematically analyze one representative and impactful ambiguity that reveals a critical gap when deploying embodied agents in real-world settings. This work is intended as an initial step—a “starting point” that highlights an important yet understudied challenge.
>
> To conduct a rigorous and controlled investigation, we deliberately focus on name–referent ambiguity, which is well-established in linguistic theory. As described in the *“Names and Referents’’* chapter of *Pragmatics* [1], this form of ambiguity frequently appears in daily communication, such as brand names, family members, rooms, or other commonly referenced entities, and often leads to misunderstanding if contextual cues are missing. Several representative examples included in REI-Bench are shown in Tab. 2.
>
> By grounding our benchmark in this specific, theoretically supported ambiguity, we can more precisely analyze how current robotic RE models fail and why such failures occur. We hope this focused and systematic study can inspire broader research into vague, underspecified, and pragmatically complex language for robotic task planning. Therefore, we intentionally do not attempt to widen the scope to all perturbation phenomena in this work.
>
> **Table 2: Categories of Ambiguous Names in REI-Dataset**
>
> | **Categories**                      | **Background Prompt Used in Generation**                                                                |
> | ----------------------------------- | ------------------------------------------------------------------------------------------------------- |
> | **Person**                          |                                                                                                         |
> | &nbsp;&nbsp;&nbsp;Family member     | Mug Carter is warm-hearted, a bit of a goofball, but deeply caring…                                     |
> | &nbsp;&nbsp;&nbsp;Friend            | Pan Williams is your longtime friend from the university…                                               |
> | &nbsp;&nbsp;&nbsp;Neighbor          | Tomato King is the friendly neighbor who has lived next door for almost ten years…                      |
> | &nbsp;&nbsp;&nbsp;Colleague         | Alarm Clock is your colleague at work, thoughtful and extremely organized…                              |
> | &nbsp;&nbsp;&nbsp;Stranger          | FireHair is someone you’ve never met before, passing by your house…                                     |
> | **Brand**                           |                                                                                                         |
> | &nbsp;&nbsp;&nbsp;Electronics brand | FridgeWave specializes in consumer electronics, focusing on smartphones, tablets, smart home devices... |
> | &nbsp;&nbsp;&nbsp;Restaurant brand  | Fork&Pan Grill is a fast-food restaurant known for quick service and affordable meals…                  |
>
> **Q1-3 (W1-3): What evidence supports the claim that vagueness is more common in non-experts (e.g., elderly or children)?**
>
> **Answer:**
> Prior studies have shown that vagueness (such as implicit REs) occurs more frequently in children [2, 3] and the elderly [4]. For example, young children often produce ambiguous expressions because their perspective-taking is still developing [2], while older adults tend to rely more on pronouns due to lexical retrieval difficulties [4]. However, there is no method to quantitatively analyze the impact of the vagueness in human instructions on embodied tasks. Therefore, we propose REI-Bench, which provides a structured pipeline to fill this gap. REI-Bench can also be applied to other forms of vagueness (e.g., language ambiguity associated with Alzheimer's disease) in addition to age-specific linguistic phenomena. We plan to collect linguistic ambiguity from diverse sources and conduct category-wise evaluations in REI-Bench 2.0, and we are currently undergoing the required Institutional Review Board (IRB) review.

---

> ### Author Response · Authors · 2025-11-20
> **Response to Reviewer huym (3/5)**
>
> **W2 & Q2: Is the claim of being the “first coreferential-vagueness planning benchmark” valid? Can the paper provide a dataset-level contrast table to show how REI-Bench differs from and improves upon existing ambiguity or coreference benchmarks?**
>
> **Answer:**
> We sincerely appreciate the reviewer’s suggestion and fully acknowledge the concern regarding the original phrasing. We have removed the overclaim and revised the statement to: “the first robot task-planning benchmark that systematically models vague REs grounded in pragmatic theory.”
>
> As suggested, we compare prior datasets in Robotics, Vision, and NLP along five dimensions: Target Tasks, Evaluation of Task Planning, Modeling of Referential Vagueness, Involvement of multi-turn context, and Statistics Information. As summarized in Tab. 3, the comparison indicates that although existing datasets address linguistic ambiguity, referring expressions, or coreference, none combine systematically controlled referential vagueness with robot task planning. Thus, the contribution of REI-Bench is to introduce an evaluation axis previously absent in prior work: systematically varying referential vagueness and contextual conditions within multi-step task planning.
>
> **Table 3: Comparison of REI-Bench with Prior Ambiguity, Coreference, and Task-Planning Benchmarks**
>
> | Dataset / Benchmark           | Task                                                                                                  | Task Planning Execution? | Systematic Vagueness Modeling? | Multi-turn Context? | Dataset / Size                                                              |
> | ----------------------------- | ----------------------------------------------------------------------------------------------------- | ------------------------ | ------------------------------ | ------------------- | --------------------------------------------------------------------------- |
> | **REI-Bench (ours)**          | Evaluating coreferential vague instructions (RE-based) for robot task planning in AI2-THOR            | ✔                        | ✔                              | ✔                   | REI-Dataset / 2.7k instructions × 9 vagueness levels                        |
> | **AmbiK [5]**                 | Ambiguous natural-language task instructions for robot planning in a kitchen environment              | ✘                        | ✔                              | ✘                   | AmbiK / 1k ambiguous instructions, 2k household tasks                       |
> | **CLARA [6]**                 | Method for LLMs to classify whether the command is certain or not                                     | ✘                        | ✔                              | ✘                   | SaGC / 105 goals, 5,222 tasks (certain, ambiguous, and infeasible included) |
> | **KNOWNO [7]**                | Framework for measuring and aligning the uncertainty of LLM-based planners                            | ✔                        | ✘                              | ✘                   | No dataset proposed                                                         |
> | **DialFRED [8]**              | Questioner-performer framework                                                                        | ✔                        | ✘                              | ✔                   | 53K task-relevant questions and answers                                     |
> | **Asking Clarifications [9]** | Clarification identification, clarification question generation, and answering for ambiguous language | ✘                        | ✔                              | ✔                   | CLAQUA / 40K dialogue words                                                 |
> | **REC benchmark [10]**        | Referring-expression comprehension on natural images                                                  | ✘                        | ✘                              | ✘                   | Ref-L4 / 45,341 annotations                                                 |
> | **WinoGrande [11]**           | Text-only commonsense coreference resolution                                                          | ✘                        | ✘                              | ✘                   | WinoGrande / 44k textual coreference cases                                  |

---

> ### Author Response · Authors · 2025-11-20
> **Response to Reviewer huym (4/5)**
>
> **W3 & Q3: Can the paper provide an explicit, reimplementable description of TOCC, such as pseudocode or a data-flow diagram of the full pipeline?**
>
> **Answer:**
> Thanks for the suggestion. For better clarification, we provide the pseudocode of TOCC below and add it in Appendix C.4. As illustrated by the pseudocode, the LLM first interprets the user’s intent and rewrites the original instruction $I$ into an explicit and clear instruction $ I_{\text{clear}} $. The planner then relies solely on $ I_{\text{clear}} $ to generate a single executable action $ a $. Broadly speaking, TOCC can be viewed as a specialized form of segmented CoT, in which the reasoning process is confined entirely to the linguistic-clarification step.
>
> ---
> **Algorithm 1: Task-Oriented Context Cognition (TOCC) for Step-Level Planning**
>
> **Input:**
> - Vague instruction $I$ containing potential implicit referring expressions
> - Rewriting template $T_{\text{TOCC}}$
> - Planning template $T_{\text{plan}}$
> - Language model $M$
> - Primitive skill set $S$
>
> **Output:**
> - Selected action $a$
>
> ---
>
> 1:  $P_{\text{TOCC}} \leftarrow \text{ComposePrompt}(T_{\text{TOCC}}, I)$
>     # Construct the TOCC rewriting query by inserting the vague instruction into the rewriting template
>
> 2:  $I_{\text{clear}} \leftarrow M(P_{\text{TOCC}})$
>     # Use the LLM to rewrite the vague instruction into an explicit, unambiguous instruction
>
> 3:  $P_{\text{plan}} \leftarrow \text{ComposePrompt}(T_{\text{plan}}, I_{\text{clear}})$
>     # Inject the rewritten instruction into the planning template
>
> 4:  $a \leftarrow \text{ConstrainedDecode}(M, P_{\text{plan}}, S)$
>     # Decode with the constraint that the output must be one skill from the set $S$
>
> 5:  **return** $a$
>
> ---
>
>
>
> **W4 & Q4: How do the token length, cost, and latency of AP, CoT, ICL, and TOCC compare? Why are stronger variants, such as gated AP and short/segmented CoT, not included? When these variants are added, does TOCC still maintain its advantage?**
>
> **Answer:**
> We appreciate the suggestions and immediately conduct extensive experiments, which compare TOCC with stronger variants of baselines regarding token length and latency. As shown in Tab. 4, our evaluation ensures fairness by assessing all methods under a unified token and latency budget. As shown in Tab. 5, TOCC adds only a minor cost compared with the light baselines, while remaining far more efficient than CoT and ICL. For instance, TOCC increases total tokens and latency by only 3.95% / 26.18% over the vanilla method and 2.38% / 22.87% over AP, but reduces them by 45.32% / 15.20% relative to Short CoT and lowers token usage by 38.41% compared with standard ICL. Tab. 5 has been incorporated into the revised manuscript in Appendix A.4.
>
> **Table 4: Token and latency limits used as the unified inference budget for each planning run**
>
> | Setting | Value |
> |--------|--------|
> | **Token limits** | 50,000 tokens |
> | **Latency limits (ms)** | 10,000 |
>
> **Table 5: Average token usage and inference latency per planning step across all planning methods.  Tested with LLaMA3.1-8B + SayCan.**
>
> | Method                | Avg Input Tokens | Avg Output Tokens | Avg Total Tokens | Avg Latency (ms) |
> |-----------------------|------------------|--------------------|-------------------|-------------------|
> | LLaMA3.1-8B + SayCan  | 1822             | 45                 | 1867              | 474.30            |
> | AP                    | 1850             | 42                 | 1892              | 492.75            |
> | Gated AP              | 1841             | 43                 | 1884              | 487.07            |
> | CoT                   | 3485             | 128                | 3613              | 917.92 (timeout observed) |
> | Short CoT             | 3464             | 82                 | 3546              | 705.64            |
> | Segmented CoT         | 2595             | 137                | 2732              | 946.36 (timeout observed) |
> | ICL (2-shot)          | 3075             | 46                 | 3121              | 514.65            |
> | **TOCC (ours)**       | **1894**         | **62**             | **1956**          | **598.40**        |

---

> ### Author Response · Authors · 2025-11-20
> **Response to Reviewer huym (5/5)**
>
> **W4 & Q4: How do the token length, cost, and latency of AP, CoT, ICL, and TOCC compare? Why are stronger variants, such as gated AP and short/segmented CoT, not included? When these variants are added, does TOCC still maintain its advantage?**
>
> **Answer:**
> In addition, we incorporate the stronger baselines suggested by the reviewer, including gated AP and short/segmented CoT (the original paper already used short CoT), and compare them with the baselines reported in the manuscript. Gated AP first classifies whether the instruction is vague and activates the aware prompt only when vagueness is detected. This design prevents the planner from hallucinating unnecessary awareness steps when the instruction is already clear. Short CoT limits the maximum number of reasoning tokens. Segmented CoT performs step-wise reasoning by generating a short chain of thought for each step rather than producing a single long sequence. We evaluated all variants using LLaMA3.1-8B + SayCan. As shown in Tab. 6, Gated AP yields only a 1.9% improvement over AP and performs comparably to the vanilla method. Similarly, both CoT and Segmented CoT underperform compared with the Short CoT variant used in our main experiments. We updated the manuscript to report the best-performing variants for each baseline (Gated AP for AP, Short CoT for CoT). The updated figures have replaced Fig. 5 and Fig. 8 in the manuscript. Details regarding the variants used for the baselines are documented in Appendix C.
>
> **Table 6: Success rates (%) of various prompting methods with variants applied to LLaMA 3.1-8B models with the SayCan framework on the REI dataset.**
>
> | Setting | LLaMA | AP | Gated AP | CoT | Short CoT | Segmented CoT | TOCC |
> |--------|:-----:|:--:|:--------:|:--:|:---------:|:--------------:|:----:|
> | Explicit REs & Standard Context | 46.9 | 40.0 | 46.8 | 42.5 | 47.3 | 43.1 | 59.0 |
> | Explicit REs & Noised Context   | 45.2 | 41.6 | 45.7 | 40.8 | 47.9 | 41.2 | 59.0 |
> | Explicit REs & Short Context    | 50.3 | 45.1 | 50.1 | 41.6 | 48.2 | 43.5 | 55.8 |
> | Mixed REs & Standard Context    | 30.1 | 29.0 | 29.0 | 26.9 | 30.9 | 24.8 | 33.6 |
> | Mixed REs & Noised Context      | 29.7 | 28.9 | 28.7 | 24.6 | 28.6 | 22.4 | 33.1 |
> | Mixed REs & Short Context       | 28.8 | 27.2 | 28.3 | 23.2 | 26.4 | 25.3 | 29.4 |
> | Implicit REs & Standard Context | 22.1 | 22.6 | 22.7 | 19.1 | 22.1 | 22.6 | 29.3 |
> | Implicit REs & Noised Context   | 21.2 | 21.4 | 21.6 | 19.4 | 20.4 | 18.2 | 29.4 |
> | Implicit REs & Short Context    | 20.8 | 21.5 | 21.2 | 20.6 | 19.0 | 18.5 | 25.1 |
>
> **References**
> ```
> [1] Levinson S C. Pragmatics[M]. Cambridge University Press, 1983.
> [2] Robinson E J, Apperly I A. Children's difficulties with partial representations in ambiguous messages and referentially opaque contexts[J]. Cognitive Development, 2001, 16(1): 595-615.
> [3] So W C, Demir Ö E, Goldin-Meadow S. When speech is ambiguous, gesture steps in: Sensitivity to discourse-pragmatic principles in early childhood[J]. Applied psycholinguistics, 2010, 31(1): 209-224.
> [4] Hendriks P, Koster C, Hoeks J C J. Referential choice across the lifespan: Why children and elderly adults produce ambiguous pronouns[J]. Language, cognition and neuroscience, 2014, 29(4): 391-407.
> [5] Ivanova A, Eva B, Volovikova Z, et al. AmbiK: Dataset of Ambiguous Tasks in Kitchen Environment[C]//Proceedings of the 63rd Annual Meeting of the Association for Computational Linguistics (Volume 1: Long Papers). 2025: 33216-33241.
> [6] Park J, Lim S, Lee J, et al. Clara: classifying and disambiguating user commands for reliable interactive robotic agents[J]. IEEE Robotics and Automation Letters, 2023, 9(2): 1059-1066.
> [7] Ren A Z, Dixit A, Bodrova A, et al. Robots that ask for help: Uncertainty alignment for large language model planners[J]. arXiv preprint arXiv:2307.01928, 2023.
> [8] Gao X, Gao Q, Gong R, et al. Dialfred: Dialogue-enabled agents for embodied instruction following[J]. IEEE Robotics and Automation Letters, 2022, 7(4): 10049-10056.
> [9] Xu J, Wang Y, Tang D, et al. Asking clarification questions in knowledge-based question answering[C]//Proceedings of the 2019 Conference on Empirical Methods in Natural Language Processing and the 9th International Joint Conference on Natural Language Processing (EMNLP-IJCNLP). 2019: 1618-1629.
> [10] Chen J, Wei F, Zhao J, et al. Revisiting referring expression comprehension evaluation in the era of large multimodal models[C]//Proceedings of the Computer Vision and Pattern Recognition Conference. 2025: 513-524.
> [11] Sakaguchi K, Bras R L, Bhagavatula C, et al. Winogrande: An adversarial Winograd schema challenge at scale[J]. Communications of the ACM, 2021, 64(9): 99-106.
> ```

---

> > ### Comment · Reviewer_huym · 2025-11-24
> >
> > I appreciate your thorough response and the supplementary information provided. As it address most of my concerns, I would raise my score to 4.

---

> ### Author Response · Authors · 2025-11-24
> **Looking forward to your reply**
>
> Dear Reviewer huym,
>
> As the rebuttal deadline is approaching, could you please have a look at our response? We are looking forward to your reply!
>
> Feel free to let us know if you have any other concerns. Thanks for your time and effort!
>
> Best Regards,
>
> Authors of Submission 3905

---

> ### Author Response · Authors · 2025-11-24
> **Thanks for your appreciation and reply**
>
> Thank you very much for your positive update and for acknowledging that our response has addressed most of your concerns. We truly appreciate the time and effort you have invested in reviewing our work.
>
> At the same time, since a score of 4 is still below the acceptance bar, **we would be grateful to understand which concerns, if any, you feel remain unaddressed**. We want to ensure we **fully resolve them to improve the manuscript quality**, and we are highly committed to further improving the clarity and technical quality of our paper during the rebuttal window.
>
> We have uploaded **an updated manuscript incorporating all your suggestions** for your review. If you could kindly indicate any remaining issues or points that need refinement, we would be very happy to provide additional clarification or revisions.
>
> Thank you again for your careful evaluation and constructive feedback.

---

> > ### Author Response · Authors · 2025-11-26
> > **Looking forward to further feedback**
> >
> > Thank you for acknowledging that our rebuttal has addressed most of your concerns. We sincerely appreciate your time and effort.
> >
> > That said, since a score of 4 remains below the acceptance threshold, **we would really appreciate knowing which specific concerns you still find unresolved**. Please feel free to check our updated manuscript, which integrates all your suggestions. We are fully committed to addressing any remaining issues during the rebuttal window.
> >
> > If there are points that continue to prevent the paper from meeting the acceptance bar, please kindly let us know. **We would be more than willing to clarify or revise further**.
> >
> > Thank you again for your evaluation and feedback.

---

> > > ### Comment · Reviewer_huym · 2025-11-26
> > >
> > > Thank you for the substantial additional experiments and clarifications provided during the rebuttal. My updated score reflects the improvements; however, it is ultimately based on my assessment of the benchmark’s overall contribution and the methodological novelty. Therefore, I will maintain the current score.

---

> ### Author Response · Authors · 2025-11-28
> **On Problem-Centric Novelty: Why Our Contribution Is Substantial and Timely**
>
> Dear Reviewer #huym,
>
> We sincerely appreciate your comments and the time you devoted to reviewing our work.
>
> Regarding your comment on the “overall contribution and the methodological novelty,” we respectfully note that our primary contribution does not lie in proposing a complex algorithmic advancement. Instead, our central contribution is the **formulation of an overlooked and practically important problem**: *referential vagueness in human-robot interaction (HRI)*, studied by an *interdisciplinary view of HRI and linguistics*. This problem is fundamental to real-world embodied AI systems but has not been systematically studied in prior literature. Our benchmark offers rigorous characterization, empirical evidence, and a simple yet effective baseline, providing meaningful value to the HRI and Embodied AI community from both robotics and linguistics perspectives.
>
> We would like to emphasize that **problem-centric benchmarks with limited scope are common and highly impactful**, even without complicated algorithmic designs. For example,
>
> * *Dissecting Dissonance* [1] focuses on self-contradictory prompts within a specific scope and a simple yet effective baseline, receiving significant attention;
> * *MM-SafetyBench* [2] investigates multimodal adversarial prompts in a similarly narrow but crucial scope;
> * *LoTa-Bench* [3] systematically benchmarks the performance of language-oriented task planners for embodied agents without proposing a new approach.
>
> In line with these works, our benchmark **identifies, formalizes, and quantitatively evaluates a critical failure mode** that modern embodied agents, i.e., LLM-, VLM-, and VLA-driven systems, frequently encounter during real-world deployment. **Given the community’s rapid adoption of end-to-end large models in robotics, we believe our findings serve as a timely and important warning signal to ensure robustness and trustworthy behavior in human-facing applications.**
>
> If you have any remaining *specific* concerns about the contribution or novelty, we would be very grateful if you could let us know and initiate some fruitful discussions. We are highly willing to improve the paper quality by clarifying or further refining our manuscript. Provided that the submission does not exhibit any flaws, we respectfully expect that it be assessed on its actual merit in a manner consistent with ICLR’s established review standards.
>
> Thank you so much for your time and effort!
>
> Best regards,
>
> Authors of Submission 3905
>
> **References**
>
> [1] Gao J, Gan L, Li Y, et al. *Dissecting Dissonance: Benchmarking Large Multimodal Models Against Self-Contradictory Instructions.* ECCV 2024.
>
> [2] Liu X, Zhu Y, Gu J, et al. *MM-SafetyBench: A Benchmark for Safety Evaluation of Multimodal Large Language Models.* ECCV 2024.
>
> [3] Sun S., et al. *LoTa-Bench: Benchmarking Language-oriented Task Planners for Embodied Agents.* ICLR 2024

---

### Official Review · Reviewer_7zBe · 2025-10-29

**Soundness:** 3
**Presentation:** 3
**Contribution:** 2
**Rating:** 4
**Confidence:** 3

**Summary:**

This paper introduces REI-Bench, a benchmark designed to isolate and analyze how vagueness in referring expressions within human instructions affects LLM-based robot task planners. The authors automatically construct the dataset based on ALFRED, defining three levels of referential-expression difficulty and three contextual conditions. They evaluate six 7B–8B LLMs using two task planners, SayCan and LLM+P, and further investigate the causes of performance degradation under vague instructions. To mitigate the issue, they propose a simple prompting strategy, task-oriented context cognition, which decouples instruction understanding from plan generation. Overall, REI-Bench frames embodied task planning as a language understanding challenge and provides a controlled, reproducible framework for studying it.

**Strengths:**

* Highly practical research: Because people naturally use referring expressions, robots and embodied agents that coexist with humans must be able to interpret and resolve such expressions to ensure safe, natural, and efficient collaboration. In this context, REI-Bench provides a crucial foundation for evaluating whether embodied agents can truly comprehend human-like referring language which is a core capability required for real-world human–robot interaction.
* Appropriate benchmark design: The 3×3 structure of REI-Bench combining three levels of referring-expression difficulty with three context types enables a clear and fine-grained analysis of language understanding capabilities across varying degrees of linguistic vagueness, contextual noise, and dialogue-memory constraints.
* Transparent descriptions: The authors provide comprehensive documentation of the dataset construction process, detailing all prompts, constraints, and post-processing rules in Appendix B. Their three-stage pipeline comprising context generation, context processing, and referring-expression replacement ensures both reproducibility and systematic control.
* Multi-faceted experiments and analyses: The experiments and analyses are well structured, clearly showcasing the strengths of REI-Bench. By combining two planning frameworks and six LLMs, the study provides a comprehensive factorial analysis with detailed results for each condition and accompanying ablation studies. The findings are visualized across referring-expression levels and context variations, supplemented by human baselines and an interpretable taxonomy of errors, including object omission and execution failure.

**Weaknesses:**

* Potentially biased seed instructions: The dataset’s seed instructions were created only from tasks that LLaMA 3.1-8B + SayCan successfully executed in AI2-THOR. Tasks such as Pick Two & Place were simply excluded as "not reliably completed." This means the benchmark originates exclusively from simpler, one-object, short-horizon tasks, producing an inherent bias toward language situations that small models already handle. Statistics of instructions per ALFRED task types need to be presented.
* Outdated planners: Both SayCan and LLM+P represent early-generation LLM-based planning frameworks. The evaluation excludes more recent paradigms that incorporate elaborative reasoning and context management, which could have yielded more informative and comparative experimental results. Consequently, the paper’s conclusions about the “limits of LLM-based planning” may not fully generalize to modern planning agents.
* Tested with small LLMs only: All models evaluated in the experiments are small-scale LLMs with 7B - 8B parameters. Such models are known to exhibit unstable discourse reasoning and weak coreference resolution. Consequently, the sharp performance drop from explicit to implicit referring expressions may reflect model capacity limitations rather than the intrinsic difficulty of the benchmark itself. The absence of recent larger models prevents any scaling analysis that could reveal capacity thresholds for robust language understanding.

**Questions:**

Please refer to the weaknesses and address the issues.

---

> ### Author Response · Authors · 2025-11-20
> **Response to Reviewer 7zBe (1/3)**
>
> We sincerely thank Reviewer **7zBe** for the thoughtful and constructive comments, which help us improve the paper quality significantly. The reviewer’s suggestions regarding **incorporating modern frameworks and LLM models** inspire us to make our evaluation more comprehensive. We are also encouraged by the reviewer’s recognition of the value of our empirical findings, **“which help fill a gap in LLM-based robot task planning”**.
>
> In the following, we address all concerns raised by the reviewer and provide further analyses, statistics, and clarifications to strengthen the manuscript. We hope the responses can address the concerns. In addition, we have submitted a revised manuscript with an appendix where we mark all the suggested tables, figures, and analytics in magenta color.
>
> **Q1 (W1): The dataset is constructed from simpler, one-object, short-horizon tasks, which may introduce a bias toward language situations that small models already handle. The distribution of instructions across ALFRED task types should therefore be reported.**
>
> **Answer:**
> Our motivation is to investigate how implicit REs affect the planner's performance. To isolate this effect, we adopt a controlled-variable approach: we select tasks where models can reliably complete the original clear instructions, and then introduce RE instructions to measure the resulting performance drop. This setup enables a focused and systematic analysis of referential vagueness.
>
> We acknowledge that this selection introduces bias by excluding complex, long-horizon tasks. However, failures on such tasks may stem from multiple confounding factors, such as general model limitations unrelated to RE, making them unsuitable for isolating the specific impact of referential vagueness. Therefore, restricting the analysis to reliably solvable tasks represents a necessary first step, and we explicitly document this limitation in the manuscript.
>
> As future LLM-based planners become capable of solving more complex, clear-instruction tasks, we plan to extend our analysis to long-horizon scenarios in REI-Bench . In summary, this work highlights an overlooked challenge, provides a scientifically grounded analysis, and aims to motivate further research in trustworthy human–robot interaction.
>
> As suggested, we present the statistics of instructions per ALFRED task in Tab. 1. Using stratified sampling, we ensure that no task-distribution bias is introduced in REI-Bench. Tab. 1 has been added to Appendix A.2, and the corresponding revisions have been incorporated into Section 4.1 Experiment Setup in the manuscript.
>
> **Table 1: Task-type distribution and average subtask steps in the 1,000-task sampled subset. The column “Original Proportion” corresponds to the original ALFRED dataset distribution,  and our stratified sampling preserves this proportion.**
>
> | Task Type           | Original Proportion (%) | Sampled Count | Avg. Subtask Steps |
> | ------------------- | ----------------------- | ------------- | ----------------------- |
> | Cool & Place        | 16.8%                   | 168           | 12                      |
> | Heat & Place        | 16.8%                   | 168           | 14                      |
> | Clean & Place       | 16.2%                   | 162           | 10                      |
> | Examine in Light    | 13.3%                   | 133           | 4                       |
> | Stack & Place       | 18.4%                   | 184           | 11                      |
> | Pick & Place        | 18.5%                   | 185           | 9                       |
> | Pick Two (Excluded) | ---                     | 0             | ---                     |
> | **Total / Average** | **100%**                | **1000**      | **10 (Average)**        |

---

> ### Author Response · Authors · 2025-11-20
> **Response to Reviewer 7zBe (2/3)**
>
> **Q2 (W2): Can we include more recent planning paradigms that go beyond SayCan and LLM+P? Do modern paradigms with stronger reasoning or context-management capabilities address the vagueness problem?**
>
> **Answer:**
> Yes, this is a great suggestion. We have added extensive experiments using the more recent LLM-based planning frameworks, hierarchical task planning and execution (HPE) [1] and DAG-Plan [2]. *HPE* constructs a memory bank through an LLM to store key contextual information, enabling monologue-style reasoning and more coherent context management. *DAG-Plan* leverages an LLM to generate a Directed Acyclic Graph (DAG) of sub-tasks, explicitly modeling the dependencies between sub-objectives in a task.
>
> However, we find that these modern paradigms, especially HPE, may further hinder LLMs from interpreting implicit REs because their context-management mechanisms are more restrictive. As shown in Tab. 2, “LLaMA3.1-8B+HPE” exhibits a drop of 36.4% / 37.3% / 38.7% at the “Mixed REs” level and a further decrease of 6.7% / 5.4% / 4.4% under “Implicit REs.” Moreover, “LLaMA3.1-8B+DAG-Plan” shows a drop of 5.7% / 7.3% / 8.7% at “Mixed REs,” followed by an additional decrease of 9.9% / 6.9% / 6.7% at “Implicit REs.” The results demonstrate that vague instruction issues highlighted by REI-Bench are a common problem for existing LLM-based task planning agents. In the final camera-ready version, we plan to add 12 planners in total (2 frameworks × 6 LLMs) and have completed 6 of them. We will supplement other results in the subsequent discussion period and add them to Fig.4 and Appendix A.3 in the manuscript.
>
> **Table 2: Success Rate (%) of LLaMA3.1-8B under HPE and DAG-Plan across different RE types and context.**
>
> |                  |              | LLaMA3.1-8B+HPE |              |              | LLaMA3.1-8B+DAG-Plan |              |
> | ---------------- | :----------: | :-------------: |:----------: | :-------------: |:----------: | :-------------: |
> |                  | Explicit REs | Mixed REs | Implicit REs | Explicit REs | Mixed REs | Implicit REs |
> | Standard Context | 58.4 | 22.0 | 15.3 | 37.4 | 31.7 | 21.8 |
> | Noised Context   | 57.9 | 20.6 | 15.2 | 36.8 | 29.5 | 22.6 |
> | Short Context    | 59.8 | 21.1 | 16.7 | 38.7 | 30.0 | 23.3 |
>
>  **References**
> ```
> [1] Han S, Qiu B, Liao Y, et al. RoboCerebra: A Large-scale Benchmark for Long-horizon Robotic Manipulation Evaluation[J]. arXiv preprint arXiv:2506.06677, 2025.
> [2] Gao Z, Mu Y, Qu J, et al. Dag-plan: Generating directed acyclic dependency graphs for dual-arm cooperative planning[J]. arXiv preprint arXiv:2406.09953, 2024.
> ```

---

> ### Author Response · Authors · 2025-11-20
> **Response to Reviewer 7zBe (3/3)**
>
> **Q3 (W3): Why don’t we consider large-scale models? Can large-scale models solve the vague instruction problem?**
>
> **Answer:**
> We primarily consider small open-source models for two reasons. First, in real-world robotic settings, such as household robots, models must operate under strict computational constraints, making small-scale models more practical for deployment. Second, models intended for on-robot deployment typically require further adaptation, such as fine-tuning or reinforcement learning, which necessitates the use of open-source models. These points have been incorporated into the Introduction and Section 4.1 of the manuscript.
>
> Nevertheless, to provide a more complete benchmark comparison, we include additional experiments. As shown in Tab. 3, the success rate of “GPT-4o + SayCan” improves by an average of 32.0% over “GPT-4o-mini” but remains 28.0% below the human baseline on average. This suggests that larger models cope with vague language more effectively, while still leaving considerable room for improvement compared to human performance. Tab. 3 has been incorporated into Fig. 4 of the revised manuscript.
>
> **Table 3: Success rates (%) of GPT-4o and GPT-4o-mini using the SayCan framework, compared with the human baseline.  Results under three types of contexts for three types of REs are reported.**
>
> |                      |                  | GPT-4o-mini + SayCan |                  |                  | GPT-4o + SayCan |                  |                  | Human Baseline |                  |
> | -------------------- | :--------------: | :------------------: | :--------------: | :--------------: | :-------------: | :--------------: | :--------------: | :------------: | :--------------: |
> |                      | **Explicit REs** |    **Mixed REs**     | **Implicit REs** | **Explicit REs** |  **Mixed REs**  | **Implicit REs** | **Explicit REs** | **Mixed REs**  | **Implicit REs** |
> | **Standard Context** |      45.00       |        25.90         |      24.30       |      73.50       |      61.60      |      52.60       |       97.0       |      91.0      |       86.0       |
> | **Noised Context**   |      43.10       |        26.20         |      21.30       |      72.90       |      61.20      |      52.40       |       95.0       |      92.0      |       85.0       |
> | **Short Context**    |      44.40       |        23.30         |      21.30       |      73.10       |      61.80      |      52.90       |       98.0       |      91.0      |       79.0       |

---

> ### Author Response · Authors · 2025-11-24
> **Looking forward to your reply**
>
> Dear Reviewer 7zBe,
>
> As the rebuttal deadline is approaching, could you please have a look at our response? We are looking forward to your reply!
>
> Feel free to let us know if you have any other concerns. Thanks for your time and effort!
>
> Best Regards,
>
> Authors of Submission 3905

---

> > ### Comment · Reviewer_7zBe · 2025-11-25
> > **Thank you for the responses with additional experimental results**
> >
> > I appreciate the authors' extensive responses with the supplementary experimental results. These addressed most of my doubts and questions. I would raise my score to 6.

---

> > > ### Author Response · Authors · 2025-11-25
> > > **Appreciation for the constructive comments**
> > >
> > > We sincerely appreciate your helpful suggestions! Thanks for your time and effort!

---

### Author Response · Authors · 2025-11-20
**Summary of the rebuttal and the major changes of the revised manuscript**

Dear reviewers,

We would like to express our heartfelt gratitude for your invaluable time, expertise, and meticulous attention in reviewing our manuscript. The insightful comments and constructive feedback have inspired us to work diligently to address all inquiries by conducting extensive experiments and clarifications, which immensely enrich the quality and rigor of our work.

We appreciate that the reviewers acknowledge the advantages of our work: includes **“highly practical research, appropriate benchmark design, and multi-faceted experiments and analyses”** (reviewer 7zBe), **“clear problem framing motivation, and salient empirical findings”** (reviewer huym), and **“a novel benchmark, a crucial problem, and extensive experiments”** (reviewer YMUn).

Apart from responses, a **revised manuscript** has been submitted, accompanied by an appendix that delineates all revisions, highlighted in magenta color for clarity. Owing to space constraints, selected experiments have been incorporated into the main manuscript, while supplementary experiments have been included in the appendix. Allow me to summarize the significant alterations made in both the rebuttal and the revised manuscript:

- **Expanded Evaluation Across Planning Frameworks:**
In addition to SayCan and LLM+P, we add two advanced planning frameworks, HPE and DAG-Plan, to the benchmark, which provide a more comprehensive evaluation across diverse modern planning frameworks. Additional results have been added to Fig. 4, and the remaining results will be updated in Fig. 7 during the discussion period.

- **Enhanced Baseline with Variants:**
We enhance the core baselines by introducing variants and improving prompt designs to ensure a fair and competitive comparison. Updated results have been added to Fig. 5, Fig. 8, Tab. 3, and Tab.6.

- **Extended Benchmark Comparison:**
We add a comparison table that contrasts REI-Bench with prior datasets and methods involving ambiguity, coreference, and robot task planning, highlighting the unique evaluation dimension introduced by our benchmark. The comparison has been added to Tab. 1.

- **Refined TOCC Description and Standardized Pseudocode:**
We refine the TOCC description and provide standardized pseudocode in Appendix C.4 to improve clarity, reproducibility, and conceptual grounding.

- **Quantitative Analysis of Task-Type Distributions:**
We compare task-type distributions before and after sampling in Tab. 5, addressing concerns regarding potential selection bias.

- **Supplementary Evaluation with Large-Scale LLMs:**
We include results from a large-scale model, GPT-4o, to compare its performance with the smaller models suitable for on-robot deployment. Additional results have been added to Fig. 4.

We believe that the revisions made to the manuscript, along with the detailed rebuttals, have significantly improved its quality and addressed the concerns raised, making it more satisfactory for publication.

Best Regards,

Authors of Submission 3905

---

### Meta-Review · Area_Chair_vTJn · 2026-01-04

**Summary:**

Reviewers generally found the problem important and the benchmark design well-motivated, but raised concerns about (i) potential bias in benchmark construction due to filtering tasks solvable by small models, (ii) limited coverage of modern planning frameworks in the initial submission, (iii) reliance on small LLMs without scaling analysis, and (iv) whether the contribution—particularly TOCC—offered sufficient methodological novelty beyond a diagnostic benchmark. Some reviewers also questioned the generality of conclusions beyond coreferential vagueness and short-horizon tasks.

**Reviewer Concerns:**

Addressed concerns:

- Planner coverage: The authors substantially expanded the evaluation to include three planning frameworks (SayCan, DAG-Plan, HPE) and six LLMs, resulting in 12 planners in total, directly addressing concerns about outdated or narrow planner choices.

- Benchmark bias and task distribution: The revised manuscript includes task-type distribution statistics before and after sampling, clarifying the nature and impact of the task filtering process.

-  Model scale: The authors added results with a larger model (GPT-4o) alongside small open-weight models, helping contextualize whether observed failures are due to linguistic difficulty rather than model capacity alone.

- Clarity and reproducibility: The dataset construction, evaluation protocol, and TOCC method were clarified, with added pseudocode and comparison tables situating REI-Bench relative to prior benchmarks.

Outstanding concerns:

- Scope of linguistic vagueness: The benchmark primarily targets coreferential vagueness, and does not yet cover other forms of linguistic ambiguity (e.g., descriptive or pragmatic vagueness).

- Task complexity: By design, the benchmark excludes longer-horizon or multi-object tasks that are not reliably solvable by current planners, limiting generalization to more complex embodied settings.

- Methodological novelty: TOCC is intentionally simple and functions more as a mitigation baseline than a novel planning algorithm; some reviewers remained unconvinced that this aspect alone constitutes a strong standalone contribution.

These limitations are acknowledged in the revised manuscript and framed as scope constraints.

**Reviewer Scores:**

All reviewers engaged in the discussion.

- Reviewer 7zBe: The reviewer explicitly states in the discussion that the supplementary experimental results “addressed most of my doubts and questions” and that they “would raise my score to 6.”

- Reviewer huym: The reviewer explicitly states: “As it addresses most of my concerns, I would raise my score to 4.”  In a later follow-up, huym clarifies that while improvements were acknowledged, the (updated) score remains driven by their view of benchmark contribution and methodological novelty, and they “will maintain the current score.”

- Reviewer YMUn: The reviewer explicitly states: “The authors have addressed most of my concerns, and I will keep my current rating,” which was already 6.

---

### Decision · Program_Chairs · 2026-01-26

Accept (Poster)